# Sea temperature and pollution are associated with infectious disease mortality in short-beaked common dolphins
Rosie S. Williams [1,2] ✉, David J. Curnick [1], Andrew Baillie[3], Jonathan L. Barber[4], James Barnett[5], Andrew Brownlow [6], Robert Deaville[1], Nicholas J. Davison[6], Mariel ten Doeschate [6], Paul D. Jepson, Sinéad Murphy[7], Rod Penrose[8], Matthew Perkins[1], Simon Spiro[1], Ruth Williams[9], Michael J. Williamson [1], Andrew A. Cunningham [1] & Andrew C. Johnson [10]

The concurrent pressures of climate change and chemical pollution, often studied in isolation, have been linked to increases in infectious disease that threaten biodiversity. Understanding their interconnected nature is vital, as the impacts of climate-mediated environmental changes can be exacerbated by chemical pollution and vice versa. Using data from 836 UK-stranded short-beaked common dolphins (*Delphinus delphis*) (n = 153 (analysed for polychlorinated biphenyl (PCB) blubber concentrations)) necropsied between 1990 and 2020, we show that PCB concentrations and sea surface temperatures (SSTs) are associated with an increased risk of infectious disease mortality. Specifically, a 1 mg/kg lipid increase in PCB concentration correlates with a 1.6% increase in disease mortality risk, while a 1 °C rise in SST corresponds to a 14% increase. Additionally, we derived a novel PCB threshold concentration (22 mg/kg lipid), defined as the level where PCB blubber concentrations are significantly associated with infectious disease mortality risk. International efforts to reduce carbon emissions have mostly failed, and despite regulatory efforts, PCBs remain a significant threat. We demonstrate the urgent need for conservation strategies that address both risk factors simultaneously to protect marine biodiversity.

Climate change and chemical pollution have both been linked to increases in infectious disease that pose a significant threat to biodiversity[1–4], especially in marine environments that often serve as the final sink for persistent chemical pollutants (POPs)[5]. The cumulative impacts of these stressors are likely to produce compounding effects. For example, POPs, such as polychlorinated biphenyls (PCBs), cause immunosuppression, which may be exacerbated by climate-mediated environmental changes[6]. Conversely, chemical pollutants may increase the susceptibility of species to the impacts of climate change[6–8]. Advancing our knowledge of how these stressors relate

to infectious disease is vital to the development of effective conservation strategies.

PCBs, which are particularly persistent and toxic in comparison to other POPs, are of particular concern as they impair reproduction and produce immunotoxicity in marine mammals[9–11]. Despite the European ban on PCBs in the mid-1980s, large amounts still require disposal, and they continue to persist in the environment due to their high stability and bioaccumulative nature[12,13]. Further, legacy PCBs continue to enter the marine environment via several mechanisms such as terrestrial runoff,

[1]Institute of Zoology, Zoological Society of London, London, United Kingdom. [2]Department of Genetics, Evolution and Environment, University College London, London, United Kingdom. [3]The Natural History Museum, London, United Kingdom. [4]Centre for Environment, Fisheries and Aquaculture Science (Cefas), Lowestoft, United Kingdom. [5]Cornwall Marine Pathology Team, Fishers Well, Higher Brill, Constantine, Falmouth, United Kingdom. [6]School of Biodiversity, One Health and Veterinary Medicine, College of Medical, Veterinary & Life Sciences University of Glasgow, Glasgow, United Kingdom. [7]Marine and Freshwater Research Centre, Department of Natural Sciences, School of Science and Computing, Atlantic Technological University, Galway, Ireland. [8]Marine Environmental Monitoring, Penwalk, Llechryd, Cardigan, United Kingdom. [9]Cornwall Wildlife Trust, Five Acres, Allet, Truro, United Kingdom. [10]UK Centre for Ecology and Hydrology, Wallingford, United Kingdom. Paul D. Jepson: Unaffiliated. ✉e-mail: rosie.williams@ioz.ac.uk; rosie.williams.16@ucl.ac.uk

inadequate waste disposal, intentional discharge, dredging, atmospheric transport, dispersion and deposition[14–17]. However, despite the general adherence to international regulatory agreements like the Stockholm Convention, tissue concentrations of PCBs remain at hazardous levels in many wildlife species[13,18].

Concurrently, climate change is precipitating rapid alterations to the marine environment, including increases in the frequency and severity of marine heatwaves[19]. In the UK, there have been increases in mean sea surface temperatures and several marine heatwaves[20–22]. These changes are having profound effects on marine ecosystems, including shifts in species distribution, changes to prey availability and altered pathogen-host dynamics[23]. Further, warming ocean temperatures have been linked to greater incidence and spread of diseases in marine species, potentially amplifying the immunosuppressive impacts of chemical pollutants like PCBs[6,24].

Marine mammals are excellent sentinels for marine ecosystem health owing to their high trophic position, long lifespan and thick layer of blubber where lipophilic pollutants accumulate[25]. They accumulate some of the highest recorded concentrations of PCBs in wildlife, and exposure to high concentrations has been associated with immune system impairment in in vivo and in vitro studies[9,26]. In the UK, short-beaked common dolphins (*Delphinus delphis*) are exposed to high levels of PCBs and shifts in their distribution have been associated with changes in sea surface temperature (SST)[18,22,27]. Moreover, the rate of decline of PCBs in the blubber of UK-stranded short-beaked common dolphins is slower compared to other odontocete species, suggesting an elevated risk of exposure and adverse effects[18,28]. In addition, shifts from pelagic to coastal waters may be occurring, as evidenced by increased near shore sightings[29–31]. This may reflect a response to climate-induced changes in prey availability, while also increasing exposure to PCBs and other pollutants as coastal environments and biota are often more heavily polluted[14,16,32]. Previous changes in short-beaked common dolphin distribution in the UK have been linked to changes in the distribution of important prey species[33]. These changes demonstrate the intricate relationship between climate-mediated environmental changes and pollutant exposure. Considering the impacts of these pressures together not only has local relevance but also global implications, given similar threats faced by marine mammals in industrialised coastal regions worldwide.

Determining the threat from stressors such as climate-mediated environmental changes and chemical pollutants is a challenging task.

Many studies have reported associations between exposure to PCBs and immunosuppression and/or infectious disease in marine mammals[9,26,34]. Further, there are several possible mechanisms whereby warming ocean temperatures could increase rates of infectious disease (e.g., increased pathogen transmission rates or reduced prey quality and availability compromising immune function). Infectious diseases represent a significant threat to health and survival, with gastritis/enteritis, meningoencephalitis and pneumonia (bacterial, viral and parasitic) being the most prevalent[35]. Gastritis and enteritis, caused by a range of infectious agents, affect the digestive tract, leading to reduced nutrient absorption, dehydration, and, in severe cases, mortality[36]. Meningoencephalitis, often associated with bacterial or viral pathogens, causes inflammation of the brain and its surrounding membranes, leading to neurological symptoms that can impact navigation, social behaviours and feeding capacity crucial for survival[37]. Pneumonia, a respiratory infection that causes inflammation of the alveoli, causes breathing difficulties, which can result in death[36,37]. While it is not possible to carry out controlled experiments, epidemiological studies can be carried out to investigate associations between infectious disease and exposure to potential stressors such as PCB concentrations and changing ocean temperatures[34,38].

Using data obtained from necropsies of stranded short-beaked common dolphins over three decades (1990–2020), we sought to understand the cumulative impacts of chemical pollution and climate-induced changes on infectious disease susceptibility in this species. By investigating temporal trends in infectious disease mortality and the possible associations between PCB blubber concentrations, SST and infectious disease mortality, we provide insights into the broader implications of these environmental stressors on the health of this sentinel species. In addition, we determine temporal trends in PCB blubber concentrations and derive a novel infectious disease mortality risk threshold for PCBs in marine mammals.

## Results
### Temporal trends in causes of mortality
When we analysed temporal trends in overall strandings of short-beaked common dolphins ($n = 3197$), we found that between 1990 and 2020 there was a significant increase in the overall number of strandings in the UK ($\tau = 0.56$, 2-sided $p$-value = $5.20 \times 10^{-5}$) (Fig. S1). When we investigated trends in causes of death, using the smaller dataset, which contains necropsy information and cause of death ($n = 836$) (Fig. 1), we found a significant

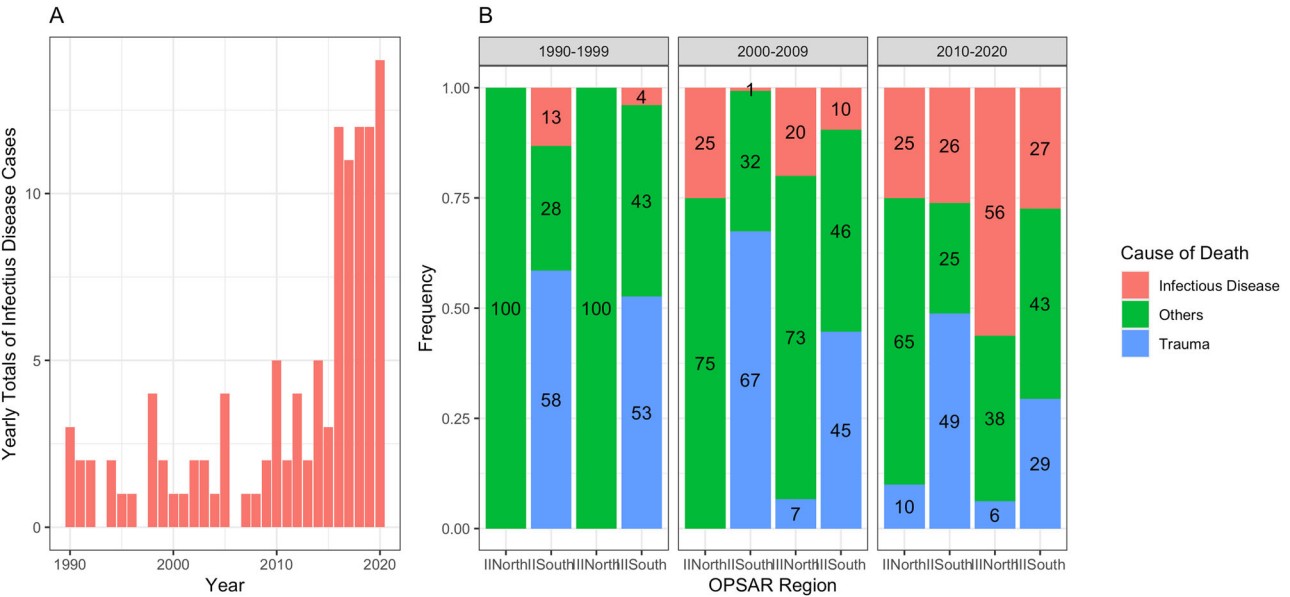

**Fig. 1 | Temporal trends in causes of mortality in UK-stranded common dolphins. A** Yearly numbers of short-beaked common dolphin infectious disease cases diagnosed at necropsy. **B** Regional proportions for each cause of death category for each decade. The figures within each bar indicate the percentage for each cause of death category. OSPAR regions represent the Convention for the Protection of the Marine Environment of the North-East Atlantic (the 'OSPAR Convention') management regions in UK waters.

increase in the number of stranded short-beaked common dolphin that died from infectious disease across all regions ($\tau = 0.48$, 2-sided *p*-value = $8.50 \times 10^{-4}$) (Fig. 1A). However, there were no statistically significant temporal trends in the other cause of death categories, when analysed over 30 years (Table S1). We were unable to test trends in specific types of infectious disease (e.g., pneumonia, gastritis, meningoencephalitis) due to the small sizes in each category (Table 2). We did plot the absolute number of cases and yearly frequencies in each category and visual inspection showed no single category drove the proportional increase in cases (Fig. S2). When we incorporated the Oslo Paris Convention (OSPAR) region classifications (either two or four geographical regions) into our analysis, we did not find any regional differences in the temporal trends of mortality causes. Nonetheless, visual representations of these data over each decade show an increase in the proportion of deaths ascribed to infectious disease in all regions (Fig. 1B). For both the regional and UK-wide models, the negative binomial regression provided a better fit compared to the *glm* with a Poisson distribution.

### Drivers of infectious disease mortality

We found that PCB blubber concentrations and monthly mean SST are associated with an increased risk of infectious disease mortality in short-beaked common dolphin (Table 1, Fig. 2). The equations of the full model that was tested and the final model are detailed below (Equations 1 and 2). We found that the exposure odds ratio for PCB blubber concentrations and infectious disease mortality was 1.016 (1.01–1.02). Hence, for a 1 mg/kg lipid increase in PCB blubber concentrations, there is an increased relative risk of death from infectious disease of 1.6%. For context, we found that the mean PCB blubber concentration was 32.15 mg/kg lipid, which equates to a 51% increase in relative risk compared to an animal with no PCB burden. The odds ratio for monthly mean SST was 1.14 (1.07–1.21), equating to a 14% increase in relative risk of death from infectious disease for each °C increase in SST.

$$Disease\ Case \sim \beta_0 + \beta_1 Nutritional\ condition + \beta_3 Latitude + \beta_4 Longitude$$
$$+ \beta_5 SST\ Mean * \beta_5 PCB\ Blubber\ Concentration + \beta_6 Age\&Sex\ Class$$
$$+ \beta_7 Season + \beta_8 SST\ Anomaly$$

*Equation 1: The full form of the logistic regression model (before the removal of some terms owing to correlation and before terms were excluded by model averaging) used to investigate relationships between SSTs (sea surface temperature), PCB (polychlorinated biphenyl) blubber concentrations and infectious disease mortality.*

$$Disease\ Case \sim \beta_0 + \beta_1 Nutritional\ condition + \beta_3 Latitude + \beta_4 Longitude$$
$$+ \beta_5 SST\ Mean * \beta_5 PCB\ Blubber\ Concentration$$

*Equation 2: The form of the averaged logistic regression model used to investigate relationships between SSTs (sea surface temperature), PCB (polychlorinated biphenyl) blubber concentrations and infectious disease mortality.*

After evaluating all possible variable combinations from the full model (Equation 1), we ranked them by their AIC values. Our final prediction model was obtained by averaging the highest-ranked models that had a ΔAIC < 2 relative to the model with the lowest AIC. The final model was based on an average across seven models. Details on variable combinations, ΔAIC and model weights are shown in Table S2. The full averaged model included, body condition, latitude, longitude and an interaction term between monthly mean SST and PCB blubber concentration (Equation 2).

### Table 1 | Model averaged coefficients for the averaged logistic regression model used to investigate relationships between SSTs (sea surface temperature), PCB (polychlorinated biphenyl) blubber concentrations and infectious disease mortality, variables were centred and scaled

|  | Estimate | Std. error | Adjusted SE | z value | Pr(>\|z\|) |
|---|---|---|---|---|---|
| (Intercept) | −6.66 | 4.61 | 4.63 | 1.44 | 0.15 |
| Latitude | 6.18 | 4.12 | 4.16 | 1.49 | 0.14 |
| **Monthly mean SST** | **1.55** | **0.77** | **0.78** | **1.99** | **<0.05\*** |
| **PCB concentration** | **0.02** | **0.01** | **0.01** | **2.11** | **<0.05\*** |
| Body condition | −0.19 | 0.21 | 0.22 | 0.87 | 0.38 |
| Longitude | 0.14 | 0.16 | 0.02 | 0.18 | 0.86 |
| Monthly mean SST: PCB concentration | 0.00 | 0.01 | 0.02 | 0.18 | 0.86 |

SST was derived for four OSPAR areas, the averaged coefficients for SST derived for two OSPAR areas are shown in Table S3. Bold and * denotes statistical significance.

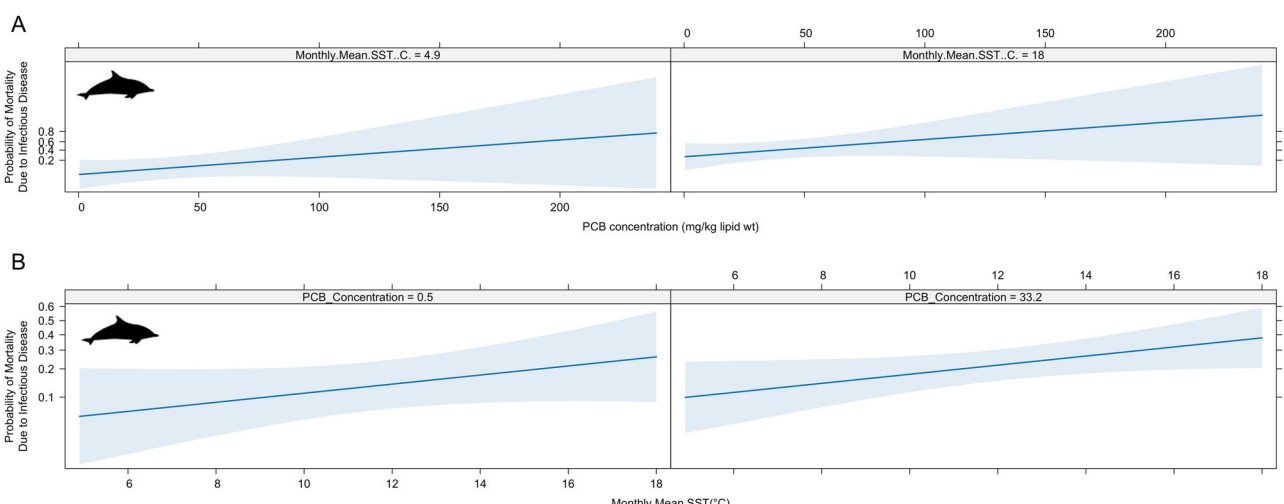

**Fig. 2 | Associations between PCB concentrations, monthly mean sea surface temperature (SST) and the probability of infectious disease mortality in UK-stranded common dolphins.** The blue line represents the probability of an individual belonging to the 'case' group (e.g., mortality due to infectious disease) against **A** PCB blubber concentrations (mg/kg lipid) shown at the minimum and maximum monthly mean SSTs in °C. **B** SST when the regions were divided in four and PCB concentration was set at the minimum and mean concentration. The shaded blue areas represent the 95% confidence intervals. The mean value has been chosen for other predictors.

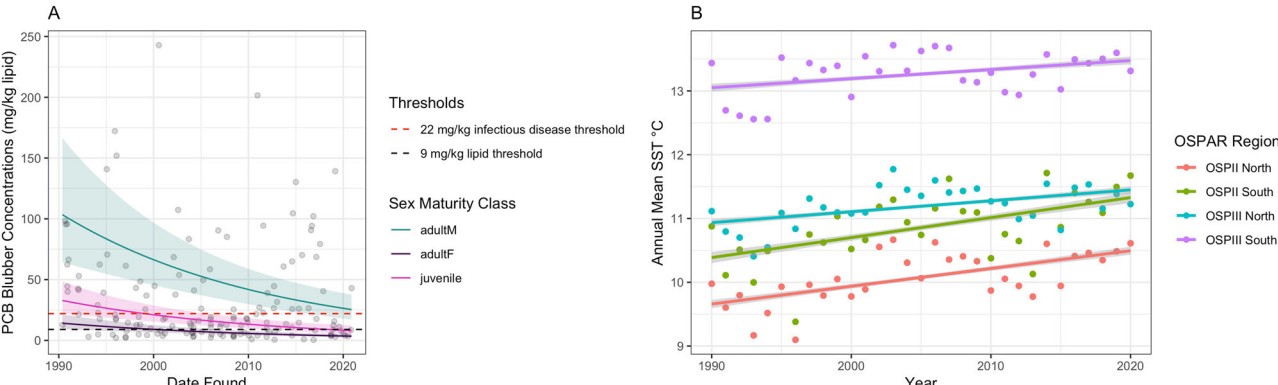

**Fig. 3 | Temporal trends in PCB blubber concentrations and sea surface temperature (SST). A** Modelled PCB blubber concentrations (mg/kg lipid) for each of the age and sex classes. The shaded ribbons represent the 95% confidence intervals. *For ease of reading, the y-axis has been limited to 150 mg/kg lipid, the full figure (without the limitation on the y-axis) can be seen in the SI Fig. 3. **B** Modelled annual mean SST ° C for each of the OSPAR regions, the increasing trend was only significant in OSPIIN and OSPIIIN. The dots represent the annual mean SST.

We found that monthly mean SST and PCB blubber concentrations were the only two significant terms in the model (Table 1).

When we included PCB concentration as a categorical variable, to derive the threshold at which PCB concentration was significantly associated with infectious disease mortality, we found this occurred at a concentration of 22 mg/kg lipid. The model equation follows the same form as Equation 2 and is detailed in the methods (Equation 4). In both cases, PCB concentration and SST were the only two significant predictors. The coefficients of the model are shown in the Table S6. To further validate the above findings, we repeated the analysis but removed infectious disease cases from the model and set live strandings as the cases with trauma and other causes of death as the controls. We found there was no significant association between live strandings, PCBs and SST (Table S7).

The results of the model with the larger necropsy dataset ($n = 836$), which did not include toxicological analysis and therefore PCBs were not included in the model, were similar in that infectious disease mortality risk was positively associated with SST and trauma and other causes of death were negatively associated with SST. The model included age and sex class, latitude, longitude, body condition and SST but only SST and latitude were significant terms. The model coefficients are shown in SI Tables 2 and 3.

### PCB blubber concentration and SST temporal trends

Modelled PCB blubber concentrations decreased between 1990 and 2020, but the rate of decline appears slow, and concentrations in adult males are still above the widely used threshold for physiological effects and the threshold we derived for increased risk of infectious disease mortality (Fig. 3A). The final model included date of stranding as the response variable with cause of death, latitude, longitude and age and sex class as explanatory variables (Table S8). When we subset the data by age and sex class we found there was no statistically significant downward trend in PCB blubber concentrations in adult males. However, there was a significant decline in PCB concentrations in adult females and juveniles (SI Tables 9–11).

When we modelled monthly mean SST we found that SSTs have significantly increased in the two northern OSPAR subregions when modelled separately (OSPIIN estimate = $9.026 \times 10^{-5}$, p-value ≤ 0.05, OSPIIIN estimate = $5.59 \times 10^{-5}$, p-value ≤ 0.05) and when modelled across the whole of the UK, including subregion as a random effect (estimate = $7.623 \times 10^{-5}$, $p < 0.05$) (Fig. 3B).

### Discussion

Here we show, for the first time in marine mammals, that PCB concentrations and SST are both positively associated with infectious disease mortality risk. Additionally, we have derived a novel PCB toxicity threshold associated with this risk, providing a crucial reference point to better inform conservation efforts. Whilst we cannot state whether the relationship between SST and infectious disease is causal, our results align with studies indicating climate change increases disease threats in multiple species. It is likely that the true driver of climate change impacts on this species is increased variability in prey availability and hence energy balance, rather than a direct physiological impact due to temperature[39,40]. Additionally, prey switching may alter pollutant exposure or have reduced nutritional value. Regardless of whether the association is causal or correlative, our temporal modelling of SSTs, coupled with projections of ongoing warming of UK waters in the coming decades[41,42] suggests this risk will increase. Our findings demonstrate an urgent need to prioritise future research to understand the mechanisms by which SST relates to infectious disease and whether this association exists in other marine megafauna and commercially important species. In contrast, the observed decline in PCB concentrations offers a cautiously optimistic prospect for decreased mortality risk, contingent upon continued efforts to prevent environmental releases and strict compliance with international agreements such as the Stockholm Convention. At present rates of elimination, several parties to the Convention will fail to meet their forthcoming commitments to eliminate the use of PCBs in existing equipment by 2025 and ensure their environmentally sound management by 2028[11,13,18,34,43,44]. Moreover, reducing the environmental levels of other immunosuppressive pollutants is critical to realising this risk reduction. It is also important to note that the modelled rate of decline of PCBs is slower in short-beaked common dolphin in comparison to other species and other persistent organic pollutants, despite similar timelines in the implementation of regulatory measures[18,34]. Given the array of global threats facing marine mammals, such as bycatch, noise pollution and prey depletion, action must be taken to reduce the impacts of chemical pollution and climate change and adopt integrative conservation strategies to mitigate the interactive nature of these threats.

It is important to acknowledge that the confounding influences on range shifts due to climate change may obscure or confound direct associations between SSTs and disease and make it difficult to identify the precise mechanisms by which SSTs affect disease dynamics. It is possible that given SSTs and rates of infectious disease mortality have increased over time, the association is reflective of an increase in another environmental variable. For instance, the association might reflect an increase in another immunosuppressive pollutant or a reduction in prey abundance or nutritional quality, both of which could weaken the immune system. A recent study on short-beaked common dolphin in the Celtic Sea reported a decline in nutritional health[45]. Further, as short-beaked common dolphin are considered to be a warm-water adapted species[46], our findings are not necessarily intuitive and may, therefore, be reflective of changes in the ecosystem precipitated by ocean warming and not mechanistically linked. However, the lack of any significant association between the live stranding cohort, PCB concentrations and SSTs does suggest that immune impairment related to these two

stressors might be a contributing factor. Given the exploratory nature of this study, further investigation is warranted to ascertain causality. Should a causal link exist, analogous to findings in other species[24,47], multiple mechanisms could explain this association. A gradual elevation of SSTs and/or extreme increases seen during recent marine heatwaves could detrimentally affect prey quality and availability, compromising immune function. For example, spatial patterns of whiting (*Merlangius merlangus*) abundance, an important prey species[48] are related to spatial patterns of SST[49]. Additionally, climate warming may increase the incidence of infectious disease through increased development rates of opportunistic pathogens that can reproduce in the marine environment and increased susceptibility and transmission rates[24]. Unfortunately, due to the wide variation in the types of infectious disease observed in our study, we were unable to determine if there was an increase in specific pathogens. However, the breadth of infectious diseases found suggests the issue is more one of a less efficient immune system rather than a particular parasite/disease becoming more abundant. Future research should focus on pathogen-specific rises, particularly parasitic diseases, as increased temperatures can boost parasite metabolism in ectothermic host species, leading to greater parasite loads in prey species and damage[50–52].

SST may serve as a proxy variable for climatic warming but other climate-induced changes such as reduced dissolved oxygen, increasing ocean acidity and salinity alterations also likely influence disease dynamics[50,53]. Moreover, shifts in host range or movement patterns, driven by temperature changes, could increase pathogen exposure. However, our findings show that any northward movement of common dolphins would result in reduced risk of diseases owing to lower SSTs and lower levels of PCBs. Reports suggest that short-beaked common dolphin, previously predominantly pelagic in the UK, increasingly frequent coastal waters, possibly responding to shifting prey distributions, which may alter pathogen exposure, for example, by increasing interspecies interactions[29–31]. Numerous studies have corroborated the correlation between rising temperatures and the incidence of infectious disease in marine species[47,54]. Nevertheless, the risk of infectious disease has not increased uniformly across all marine life in response to oceanic changes[55]. Notably, in some fish and elasmobranch populations, infection risks have diminished, a trend that appears to correlate with significant declines in populations due to anthropogenic pressures[55]. This underscores the intricate and sometimes paradoxical interactions between environmental change and disease risk within marine ecosystems.

The significant increase in the risk of infectious disease mortality associated with PCB blubber concentrations supports findings that PCBs cause immunosuppression in marine mammals[26,27,38,56,57]. Our results align with studies in harbour porpoises (*Phocoena phocoena*) that report slightly higher exposure odds ratios (1.04)[34,58]. We found that body condition was included in the averaged model but did not have a significant effect on infectious disease mortality risk. This finding has parallels with recent research in bottlenose dolphins (*Tursiops truncatus*) that found body mass index was also not a predictor of 1–2 year all cause mortality[59]. The relationship between nutritional status and infectious disease is complex, as poor nutrition can weaken the immune response, but infectious disease can also result in nutritional stress. Nutritional stress can lead to blubber loss, releasing stored PCBs from fat-rich tissues into the bloodstream, where they exert greater toxicity and heighten the risk of infection and subsequent mortality[60]. Therefore, it can be difficult to tease apart the complex relationship between PCB concentrations, nutrition and disease and should therefore, be noted as a limitation of this work.

Despite the associations reported here, it is important to note the limitations. Notably, there is a degree of uncertainty regarding certain diagnoses. For example, we cannot rule out that some animals had an underlying infection that was not detected, particularly if preservation is suboptimal. Moreover, post-mortem growth of bacteria could be mistaken for septicaemia or conversely cause sepsis to go undiagnosed. Further, in terms of the scope of this work, we were unable to investigate pathogen-specific trends or account for non-fatal infections. Nevertheless,

immunosuppression induced by PCBs tends to disrupt immune system functions broadly, increasing susceptibility to any infectious disease[9]. We also cannot rule out that selection bias in the controls may have impacted our findings. For example, animals were prioritised for PCB analyses based on their decomposition code, which could introduce bias if fresher cases are more likely to have died from disease. To assess this, we compared the proportions of causes of death between the toxicological and necropsy datasets and found no significant differences. Therefore, while we attempted to select the cases and controls independently of PCB exposure, if there are discrepancies in PCB concentrations between the control group (animals that died from trauma and are hypothesised to have lower concentrations than disease cases) and the general population, this may have skewed odds ratio estimations. However, the association between PCB exposure and infectious disease mortality in cetaceans is well documented[9,26,58,61]. It is also important to acknowledge that, given the time span of the dataset, the capacity to diagnose infectious diseases in stranded marine mammals has improved. However, similar advances have also been made in diagnosing cases of bycatch and other causes of mortality[62]. Therefore, while these improvements in diagnostic capacity may have contributed in part to the temporal increase in infectious disease cases, we think it is unlikely that the increase in infectious disease cases is solely due to improved detection. Changes in diagnostic capacity may also have altered the potential for diagnosis of certain disease processes and some infectious disease sub-categories. This should not have significantly impacted our results as we did not investigate associations with specific infectious diseases however, this is an important factor to consider if specific diseases are investigated in the future. Additionally, parasites are particularly vulnerable to overdiagnosis as they are grossly visible and relatively easy to detect. A sick animal with high parasite loads may be classified as an infectious death, but there may be underlying toxic, environmental, social, or alternative infectious causes, such as immunosuppressive morbilliviruses, contributing to the condition. Our findings highlight such complexities, demonstrating the importance of a multifactorial approach in understanding disease processes in marine mammals.

A further limitation of our study was caused by collinearity between the variables which led us to exclude season from our model. This requires careful interpretation of the link between SST and infectious disease to discern whether the positive association is likely to reflect climate warming impacts or is merely a proxy for seasonal or temporal variations in mortality causes. Fishing effort, particularly off the southwest coast, is seasonal, with most bycatch strandings occurring in winter and spring. To assess the impact of seasonality, we compared infectious disease mortality against all other mortality causes, not solely bycatch. In addition, we tested for a relationship between season and mortality causes and found no significant relationship, suggesting the observed association between SST and infectious disease mortality extends beyond seasonal bycatch fluctuations. Moreover, the absolute number of infectious disease cases has significantly increased over time, whilst there was no significant increase or decrease in other causes of death. It is also important to consider the lack of a significant interaction term between SST and PCB concentrations and whether this is a true finding or an artefact of the different temporal and biological dynamics of these variables. While SST varies seasonally and can influence short-term physiological responses, PCB accumulation in blubber represents long-term pollutant exposure, potentially masking any immediate relationship with SST.

Toxicity thresholds for PCBs and other pollutants are typically derived using toxicological data in laboratory species (e.g., European mink (*Mustela lutreola*) and murine models), serving as approximations in the absence of accurate species-specific data. Developing thresholds based on real-world environmental exposures and epidemiological modelling rather than relying on in vitro studies or surrogate species is vital to understand whether enough action is being taken to minimise risk to health. To the best of our knowledge, this is the first study to derive an infectious disease mortality toxicity threshold for PCB blubber concentrations in marine mammals, providing a crucial reference point to better inform conservation

management. The threshold we derived, whereby PCB concentration had a significant effect on disease risk, was 22 mg/kg lipid, higher than the commonly referenced threshold of 9 mg/kg lipid for physiological effects in marine mammals[63] but lower than the 41 mg/kg lipid threshold for reproductive impairment in ringed seals (*Phoca hispida*)[64]. Mean PCB concentrations in adult males were above the derived threshold, indicating elevated risk in this group. This is particularly concerning as PCB blubber concentrations in adult males have not significantly declined, unlike in adult females and juveniles.

A critical question is whether SST increases, PCB levels and their relationship with infectious disease imperil the UK short-beaked common dolphin population. The latest conservation status assessment under the EU Habitats Directive in 2019 is 'unknown' making it difficult to relate our findings to the long-term health of the population. Despite this, there is some cause for tempered optimism in light of the temporal reduction in PCB concentrations when age class and sex were modelled together. However, PCB levels remained stable in adult males, which cannot offload their pollutant burden through gestation and lactation[65]. While we have demonstrated an association between PCBs and infectious disease mortality risk, further research is necessary to establish robust toxic thresholds for chronic PCB exposure in vulnerable groups, to discern whether the relationship between SST and disease risk is consistent across species, and to develop an understanding of the underlying causal mechanisms. Our research findings not only inform the conservation management of marine mammals but also offer insights into the health of marine ecosystems globally, emphasising the pressing need to address the twin threats of chemical pollution and climate change.

## Online methods
### Sampling
Short-beaked common dolphin necropsies ($n = 836$) were carried out under the UK Cetacean Strandings Investigation Programme (CSIP) between the years 1990 and 2020 according to standard procedures for marine mammals[66,67]. Toxicological analyses of blubber samples were carried out on a subset of these individuals ($n = 213$). To ensure we could control for confounding variables in the models, only individuals whose sex, cause of death, age group, body weight, body length and location information were recorded were included in the analyses. This led to a dataset of 611 necropsied individuals, of which toxicological results were available for 153. Carcasses were prioritised for toxicological analysis according to their state of decomposition using a standardised classification system for marine mammals (Codes 2-4)[66]. Of the carcasses analysed for pollutants, 74% were classified as extremely fresh or only slightly decomposed. Carcasses were prioritised in this way to minimise the impact of changes in pollutant concentrations and lipid dispersion that are associated with decomposition[68]. We ensured that the individuals that underwent toxicological analysis were a representative sample of the strandings that occurred over the study period by testing for statistical differences in the proportions of age and sex classes (juveniles, adult females and adult males) and causes of death between the toxicological dataset and the complete strandings dataset (chi-squared age and sex class $X^2 = 3.8714$, $p > 0.05$, cause of death $X^2 = 4.496$, $p > 0.05$).

As part of the pathological investigations, several biological and life-history attributes were determined. Body length and sexual maturity status were used to categorise individuals into age and sex classes. Sexual maturity was assessed using gross gonadal size and appearance and, in a representative subset, histological evidence of spermiogenesis in male testes. Female reproductive maturity was determined by identification of one or more ovarian corpora (lutea or albicantia)[69,70]. Where gonadal tissues were not assessed, classification was based on estimates of age and average body length at sexual maturity. Females were classified as sexually mature when body length was equal to or larger than 189 cm or their age was greater than 8 years[71]. Males were classified as sexually mature when their length was equal to or larger than 207 cm or their age was greater than 11.5 years[72]. For the short-beaked common dolphin, a basic index of weight-to-length ratio is

considered an appropriate metric of body condition and is widely acknowledged as a good predictor of fitness in marine mammals[73,74]. The weight and length data for the individuals in this study followed a power relationship and so a power regression model was fitted to obtain a metric that could be used as a proxy for body condition. The residuals from the best-fit regression line were extracted and used for further modelling whereby values above the model fit represented cases in good nutrition and individuals below the line represented cases in poor body condition.

To investigate spatial variation, the latitudes and longitudes of the stranding locations of each animal were collected. Strandings were divided into one of the two Convention for the Protection of the Marine Environment of the North-East Atlantic (the 'OSPAR Convention') management regions in UK waters, OSPARII (Greater North Sea) or OSPARIII (Celtic Seas), to reflect common policy reporting in the UK[75]. To account for latitudinal differences in strandings, north and south zones within these regions were created along the 55° latitude parallel. This line was chosen as the north/south divide between the western region of the UK (OSPARIII) due to its position just north of the Solway Firth, dividing the water bodies of the Irish and Celtic Seas, and Western English Channel from the West of Scotland, Malin Shelf and North Atlantic as per Williamson et al.[22] (Fig. 4). Additionally, on inspection of these data there were no strandings near this latitude so there is a natural break in these data. All spatial analyses were carried out by classifying the animals into two (OSPAR II, OSPAR III) or four (OSPAR II N & S, OSPAR III N & S) geographical areas.

### Disease diagnosis and classification
Each animal had a responsible veterinarian who contemporaneously determined the cause of death by gross examination followed by ancillary testing (including parasitology, bacteriology, virology and histology) as required. 'Infectious disease' is a broad category used for analyses within the UK strandings programme, consisting of a number of cause of death categories of infectious origin[76]. Within the broad category, we have provided further details on classification and diagnosis for the categories that contained the highest number of cases. Animals were classified as having died from parasitic gastritis and/or enteritis based on the observed extent of parasitism in the area and whether the parasite load was thought to have contributed significantly to the death of the animal (e.g., lack of ability to absorb nutrients). Animals were categorised as having died from meningoencephalitis based on visual inspection of the brain, histopathology of the brain tissue and/or positive culture. Generalised bacterial or fungal infections were diagnosed based on positive culture and histopathological examination of relevant tissues. Animals were classified as having died from pneumonia (from parasitic, bacterial and fungal infections) based on positive culture from sampled sites, histopathology and the extent of lung function impairment derived from visual inspection. In some cases, animals will have comorbidities, therefore the pathologist will assess which were the most significant in relation to cause of death and the animals will be assigned to that category.

### PCB analyses
We used a standardised methodology to extract and preserve the blubber samples for PCB analysis[77]. Briefly, blubber samples were taken from the left side of the body1 at the caudal or cranial insertion of the dorsal fin and preserved at −20 °C[77]. The Cefas laboratory (Lowestoft) determined the concentrations of the sum of 25 individual chlorobiphenyl (CB) congeners ($\sum 25$ CBs) (on a mg kg$^{-1}$ wet weight basis) using a method that was validated following participation in the (Quality Assurance of Information for Marine Environmental Monitoring in Europe) QUASIMEME laboratory proficiency scheme and followed the recommendations of the International Council for the Exploration of the Sea (ICES)[78–81]. In cases where the congener concentrations were below the limit of quantification (<0.0003 or <0.0004 mg kg$^{-1}$ wet weight), we set the concentration at half the limit[82]. The percentage of non-detects for each congener is shown in Table S12. The numbers of the International Union of Pure and Applied Chemistry CBs congeners analysed were: 18, 28, 31, 44, 47, 49, 52, 66, 101, 105, 110, 118, 128,

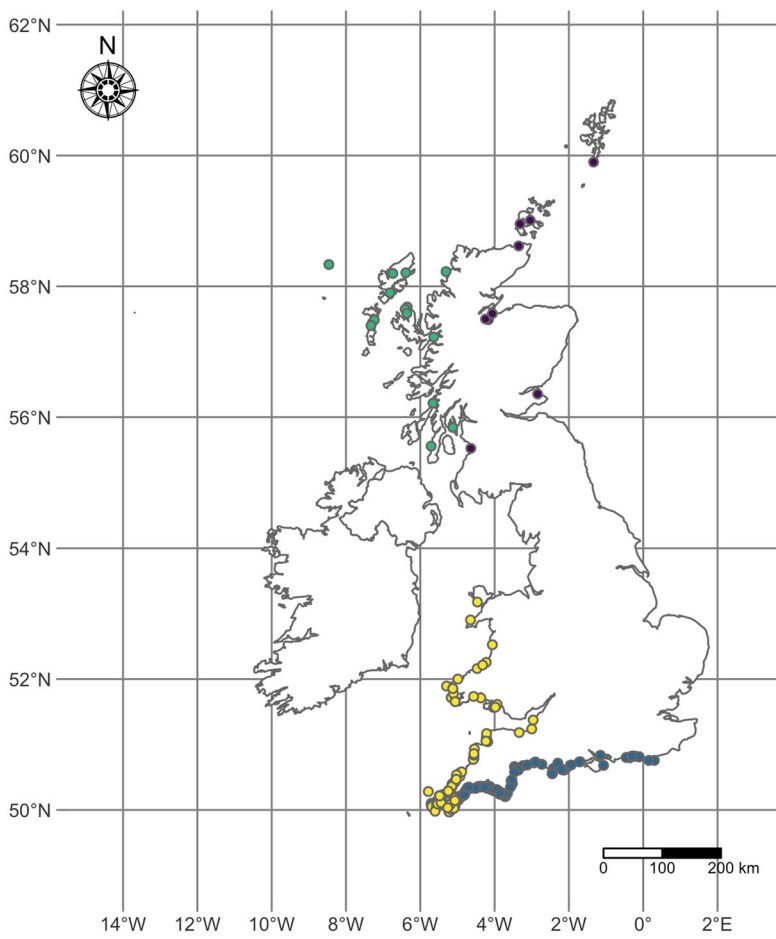

**Fig. 4 | Geographic locations and area classifications of the individuals that stranded and were included in the infectious disease mortality models (*n* = 324).** The colours of the dots represent the four regions used in the analysis. The dotted horizontal line indicates the 55° latitude parallel used as a north/south divide. The dotted vertical line indicates the border between the OSPARII and OSPARIII regions.

138, 141, 149, 151, 153, 156, 158, 170, 180, 183, 187, 194. This selection was chosen to ensure incorporation of the seven PCBs prioritised for international monitoring by ICES ($\sum$ICES7) and included those that are relatively abundant in commercial PCB mixtures with a broad range of chlorination. The sum of the 25 individual CB congener concentrations was calculated and normalised to a lipid basis (mg kg$^{-1}$ lipid) by extracting hexane from the blubber and calculating the hexane extractable lipid content[81].

The Cefas laboratory (Lowestoft) participates biannually in the QUASIMEME proficiency testing scheme for quality assurance and quality control. All the analyses were conducted under full analytical quality control procedures, including the analysis of a blank sample and certified reference material with each batch of 10 samples to assess the performance of the methods. In every case, the blanks were always below the limit of quantification. When target analytes were beyond the range of instrumentation calibration, the extracts were diluted and re-analysed. The reference material BCR349 (cod liver oil; European Bureau of Community reference) was used and the reference material results were plotted as Shewhart quality control charts for each compound. The charts were previously created from repeated analysis of the reference material using the North West Analytical Quality Analyst software™ (Northwest Analytical Inc., USA). All certified reference materials for each of the samples analysed were within the control and warning limits for each compound, defined as 2σ and 3σ – 2x and 3x the standard deviation from the mean.

**Statistics and reproducibility**

All statistical analyses were carried out using the statistical computer program R (version 4.0)[83]. Figure 5, located at the end of this section, provides further details on the datasets used in the models described below, including which variables were present in each of the subsets.

**Temporal trends in causes of mortality**. First, we tested whether there was a significant temporal trend in the overall number of reported strandings. This analysis was carried out on the wider strandings dataset (*n* = 3197) which does not include cause of death as post-mortem examinations are not carried out on the majority of stranded animals. We then tested whether there was a significant temporal trend in the number of strandings for each cause of death, using the dataset of necropsied animals (*n* = 836). Cause of death was categorised into three classes: infectious disease, trauma and others (the latter including starvation and live stranding). We also tested for trends in live standings separately as this was the second most common cause of death therefore, we wanted to investigate trends separate from the "*Others*" cause of death category. We removed mass stranding events from the live strandings so that trends were not heavily skewed by a few rare, extreme events (e.g., the mass stranding of 26 short-beaked common dolphins in 2006). Detailed causes of death for each animal and the classifications are shown in Table 2. We tested for monotonic trends using the Mann-Kendell trend test from the *Kendell* package in R.

**Infectious disease mortality, sea surface temperature and PCB concentrations**

To investigate drivers of infectious disease mortality in short-beaked common dolphin, we used a case-controlled approach to compare animals that died of infectious disease (cases) with animals that died of other causes of

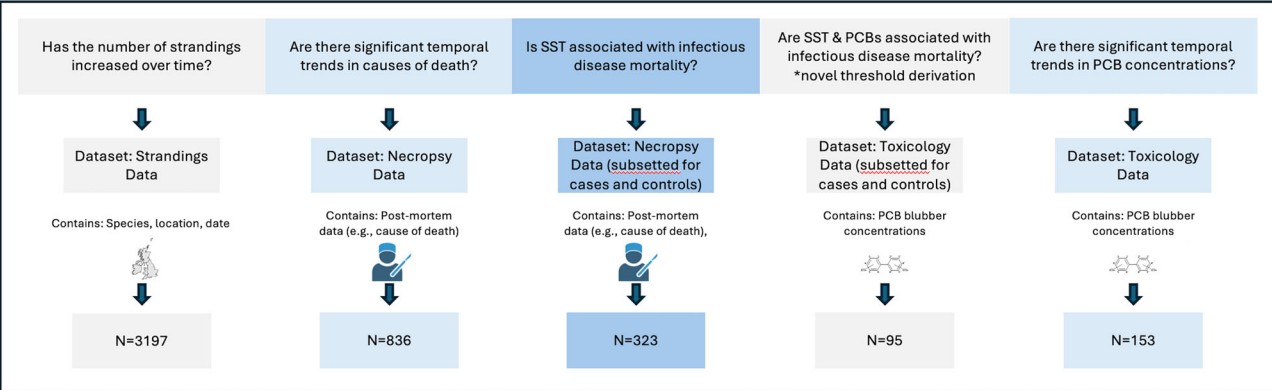

**Fig. 5 |** Schematic depicting the dataset that was used to investigate each research question and the associated sample size.

death (e.g., trauma, live stranding) (controls). First, we assessed whether infectious disease mortality was associated with SST using the larger dataset (Cases = 94, Controls = 229). Second, we assessed whether there was a relationship with SST and PCB blubber concentration using the smaller toxicological dataset (Cases = 36, Controls = 59). The complete classification of cases and controls and detailed causes of death and types of infectious disease are shown in Table 2.

We used the monthly mean SST during the month the stranding occurred as a climatic explanatory variable in the model, as it is widely considered to be a robust climate change indicator and has been shown to influence the occurrence and impact of marine infectious diseases[47,84,85]. We also included monthly SST anomaly as a climatic variable to understand if large deviations from mean SSTs may influence disease rates. For SST anomaly, the monthly mean SST values from 1990 to 2010 for each region were calculated and subtracted from the SST value for each pixel for the corresponding month, following the method by Eakin et al.[86]. Data on SST for each of the OSPAR regions were obtained from National Oceanic and Atmospheric Administration (NOAA) Climate Data Record (CDR)–Woods Hole Oceanographic Institution (WHOI): of Sea Surface Temperature Version 2[87] in Google Earth Engine[88]. For SST data, monthly means from 1990 to 2020 were calculated. The NOAA CDR WHOI SST data have a resolution of 0.25 arc degrees.

We investigated the relationship between PCB blubber concentrations, SST and infectious disease mortality by fitting generalised linear models with binomial distributions and logit link functions (logistic regression). This approach is used to predict binary outcomes, such as relative mortality risk based on various predictor variables. The probability of the outcome can then be transformed into an odds ratio, which represents how a one-unit increase in a predictor variable (e.g., SST and PCB concentration) affects the relative odds of mortality (e.g., risk of death from infectious disease). Cause of death (cases versus controls) was modelled against monthly and anomaly mean SST, PCB concentrations and other selected covariates were used as potential predictors. The potential predictors were selected according to the biological rationale that they could impact cause of death. The variables included in the full model were body condition, age and sex class, latitude, longitude, season and monthly mean SST, monthly SST anomaly, an interaction term between PCB concentrations and monthly mean SST (Equation 2). All explanatory variables were scaled and centred and assessed for multicollinearity using Variance Inflation Factors (VIF). Significant collinearity was found between season and mean SST and SST anomaly. A key aim of this study was to investigate possible relationships between SST and infectious disease mortality, therefore season and SST anomaly were removed from the model. SST anomaly was removed rather than monthly mean SST as the AIC (Akaike's Information Criterion) for the model with monthly mean SST was lower. We performed a chi-squared test to test for associations between season and case/controls to ensure monthly mean SST was not just a proxy for seasonal variations in mortality causes and found no significant differences (chi-sq. $X^2 = 5.2491$, $p < 0.05$). We tested all possible

variable combinations to obtain several candidate models which were ranked according to their AIC values ($\Delta$AIC < 2). Our final prediction model was obtained by averaging the set of plausible models (Equation 3). To validate the model, we plotted the residuals of the model against other variables and assessed the variance. We assessed the model for over-dispersion using the ratio of deviance and residual deviance (1.051) and the value was within the proposed acceptable limits outlined in the literature (<1.5)[89]. Further model validation was carried out by conducting the Hosmer Lemeshow Goodness of Fit test, which indicated a good fit[90].

$$Case \sim \beta_0 + \beta_1 Nutritional\ condition + \beta_3 Latitude + \beta_4 Longitude + \beta_5 SST\ Mean \\ * \beta_5 PCB\ Blubber\ Concentration + \beta_6 AgeSex\ Class + \beta_7 Season \\ + \beta_8 SST\ Anomaly$$

*Equation 2: The full form of the logistic regression model (before the removal of some terms owing to correlation and before terms were excluded by model averaging) used to investigate relationships between SSTs (sea surface temperature), PCB (polychlorinated biphenyl) blubber concentrations and infectious disease mortality.*

$$Case \sim \beta_0 + \beta_1 Nutritional\ condition + \beta_3 Latitude + \beta_4 Longitude \\ + \beta_5 SST\ Mean * \beta_5 PCB\ Blubber\ Concentration$$

*Equation 3: The final form of the logistic regression model (derived from Equation 2 following model averaging) used to investigate relationships between SSTs (sea surface temperature), PCB (polychlorinated biphenyl) blubber concentrations and infectious disease mortality.*

To further validate the conclusions of our model, we also investigated associations between live strandings (the second most common cause of death), PCB concentrations and SSTs. We followed the same approach as described above but excluded infectious disease cases from the analysis. Live strandings were the cases and animals that died from trauma and other causes of death were the controls.

**PCB blubber concentrations threshold derivation.** We derived a threshold at which PCB blubber concentrations have a significant effect on infectious disease mortality by altering the logistic regression model described in section 4.3.2 (Equation 2). We replaced the continuous PCB blubber concentrations with a categorical variable that indicated whether animals were above or below a certain concentration (Equation 4). We did not stratify the data by sex maturity class. We then iteratively adjusted the PCB concentration to identify the threshold at which PCB concentration reached statistical significance ($p < 0.05$) and became associated with infectious disease mortality.

$$Case \sim \beta_0 + \beta_1 Nutritional\ condition + \beta_3 Latitude + \beta_4 Longitude \\ + \beta_5 SST\ Mean * \beta_5 PCB Threshold$$

**Table 2 | Cause of death class and category for all short-beaked common dolphins that underwent necropsy**

| Cause of death class | Cause of death category | n |
|---|---|---|
| Infectious disease | Gastritis and/or enteritis | 29 |
| Infectious disease | Generalised infection, bacterial | 24 |
| Infectious disease | (Meningo) encephalitis | 18 |
| Infectious disease | Pneumonia (bacterial, parasitic and/or fungal origin) | 25 |
| Infectious disease | Peritonitis | 5 |
| Infectious disease | Pleuritis | 3 |
| Infectious disease | Generalised parasitism | 5 |
| Infectious disease | Other causes of infectious disease mortality[a] | 5 |
| Others | Live stranding | 151 |
| Others | Not established | 99 |
| Others | Starvation | 42 |
| Others | Others | 31 |
| Others | Gas embolism | 3 |
| Others | Pneumonia, non-infectious | 3 |
| Trauma | Bycatch | 346 |
| Trauma | Physical trauma, unknown aetiology | 23 |
| Trauma | Physical trauma, boat/ship strike | 12 |
| Trauma | Physical trauma, bottlenose dolphin attack | 6 |

[a]Other causes of infectious disease mortality include: Mycotic tracheitis/bronchitis/lymphadenitis, Pancreatitis, Clostridial Infection, Myocarditis, Generalised Infection, Fungal.

*Equation 4: The final form of the logistic regression model used to derive a threshold PCB concentration.*

**Temporal trends of PCB blubber concentrations & SSTs**. To examine how PCB blubber concentrations have changed over the study period, concentrations were modelled against selected covariates that may confound temporal trends. Following extensive data exploration, we established that there was a linear relationship between $\sum$25 CBs and other covariates. Therefore, we fitted several multiple linear regression models to the variables, which might explain the variability in the data using $\sum$25 CBs as the response variable. The variables included in the full model were body condition, sex maturity class, cause of death, latitude, longitude and an interaction term between age class and sex. We did not include body condition in the full model as it was closely correlated with cause of death, and when we compared the AICs of the full model including cause of death against the full model that included body condition, the model with cause of death had a lower AIC. We tested all possible variable combinations to obtain several candidate models which were ranked according to their AIC values. Our final prediction model was obtained by averaging the set of plausible models. We performed model validation by assessing the diagnostic plots, plotting the model residuals against selected variables to assess the variance and carrying out cross-validation. We also sub-setted these data by age and sex class (adult males = 38, adult females = 48, juveniles = 67) and used the same approach to model temporal trends of PCBs in each of the subsets.

To examine how mean monthly SST changed over the study period, we built a linear mixed model with monthly mean SST as the response variable, date as the explanatory variable and region as a random effect, using the 'lmer' function in the lmer4 package[91]. As linear mixed models do not provide estimates for each level of a random effect we ran linear models for each region to investigate the regional trends in monthly mean SST. Monthly mean SST was the response variable, and date was the numerical explanatory variable.

## Reporting summary

Further information on research design is available in the Nature Portfolio Reporting Summary linked to this article.

## Data availability

Raw data supporting the results are available from the from the British Oceanographic Data Centre (BODC) https://www.bodc.ac.uk/data/published_data_library/catalogue/10.5285/2e698a2f-a288-70a5-e063-7086abc072aa/[92].

## Code availability

The R code used for analyses are available from the Zenodo Digital Repository: https://doi.org/10.5281/zenodo.14871062[93].

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

## Acknowledgements

The Cetacean Strandings Investigation Programme (CSIP) coordinates the investigation of strandings in England and Wales and is funded by the Department for Environment, Food and Rural Affairs and the Devolved Government of Wales, as part of the UK Government's commitment to a number of international conservation agreements. The Scottish Marine Animal Strandings Scheme (SMASS) coordinates the investigation of strandings in Scotland under contract to the Scottish Government Marine Directorate. The authors would like to thank funders of both stranding schemes and would also like to thank the Joint Nature Conservation Committee and other members of the CSIP Project Steering Group (PSG) for their support of the UK strandings programmes. Some of the contaminant analysis was carried out under the ChemPOP project (NE/S000127/1). Life history and some pollutant data were funded by the Marie Curie International Outgoing Fellowship within the Seventh European Community Framework

Programme (Project Cetacean Stressors, PIOF-GA-2010-276145 to S.M. and P.D.J.). Additional funding was provided through the Agreement on the Conservation of Small Cetaceans of the Baltic, North East Atlantic, Irish and North Seas (ASCOBANS) (Grants SSFA/2008 and SSFA/ASCOBANS/2010/5 to S.M.) and funding awarded to Simon Northridge in SMRU—EC-funded NECESSITY Project (NEphrops and CEtacean Species Selection Information and TechnologY, contract 501605). The authors would also like to thank the Centre for Environment, Fisheries and Environmental Sciences for carrying out the chemical analysis and Defra for funding this analysis under a service-level agreement. The authors thank ASCOBANS (Agreement on the Conservation of Small Cetaceans of the Baltic, North East Atlantic, Irish and North Seas) and Natural Resources Wales. The authors would also like to thank Fiona Reid, the Sea Mammal Research Unit, and the Natural History Museum for their help in verifying the classification of a number of age classes and Dina Sadykova for advice with the statistical analysis. The authors also wish to thank the volunteers of the Cornwall Wildlife Trust Marine Strandings Network for their help retrieving carcasses and staff at what was (AH)VLA Polwhele and volunteers at Cornwall Marine Pathology Team for their assistance with post-mortem examination and sampling in Cornwall. R.W. was funded by the Natural Environment Research Council (NERC) grant NE/L002485/1 and grant NE/S000100/1 supporting the ChemPOP project. R.W., D.J.C., R.D., M.P., and M.J.W. were partially funded by Research England. The authors would also like to thank the reviewers for their valuable comments and suggestions, which greatly improved the quality of the manuscript. This work is dedicated to R.S.W.'s son, Luke, who brightens every day.

## Author contributions

Rosie S. Williams: Conceptualisation, Methodology, Writing—original draft, Formal analysis, Visualisation. David J. Currick: Methodology, Formal analysis, Writing—review & editing. Andrew Baillie: Data curation, Writing—review & editing. Jonathan L. Barber: Data curation, Writing—review & editing. James Barnett: Data curation, Writing—review & editing. Andrew Brownlow: Data curation, Writing—review & editing. Nicholas J. Davison: Data curation, Writing—review & editing. Robert Deaville: Conceptualisation, Methodology, Data curation, Writing—review & editing. Mariel ten Doeschate: Data curation, Writing—review & editing. Paul D. Jepson: Data curation. Sinéad Murphy: Data curation, Writing—review & editing. Rod Penrose: Data curation, Writing—review & editing. Matthew Perkins: Data curation, Writing—review & editing. Simon Spiro: Data curation, Writing—review & editing. Ruth Williams: Data curation, Writing—review & editing. Michael J. Williamson: Methodology, Formal analysis, Writing—review & editing. Andrew A. Cunningham: Methodology, Formal analysis, Writing—review & editing. Andrew C. Johnson: Conceptualisation, Methodology, Formal analysis, Writing—review & editing.

## Competing interests

The authors declare no competing interests.
