## [Transparent Peer Review file · Communications Biology]

Sea temperature and pollution are associated with increased risk of infectious disease mortality in short-beaked common dolphins (*Delphinus delphis*)

Corresponding Author: Ms Rosie Williams

Version 0:

Reviewer comments:

Reviewer #1

(Remarks to the Author)

Thank you for the opportunity to review this important manuscript. The major findings of the paper are pivotal in marine mammal conservation however there are several questions outstanding regarding the methods and approach prior to accepting the current conclusions.

Primarily my biggest concern with the claim of association between PCBs, SST and increased risk of death from infectious disease is the definition and classification of infectious disease. This is fundamental to the conclusion of the paper yet has been overlooked throughout the introduction, methods and discussion. I suggest to have background information in relation to the infectious diseases of concern in this population in the introduction, clear definition as to how cases were classified into the infectious disease categories and what these categories were in the methods and subsequent discussion as to how these criteria and classification could have influenced results. As it stands the reader has to go to the supplementary information to look into the table as to what the types of infectious disease are. There is no discussion as to the increase in number of cases could have been linked to increased reporting or additional diagnostic screening or due to the increased number of stranded cases in general (Supplementary Fig. 1A). I suggest that there should be additional discussion as to why the number of infectious disease cases increased. This is really fundamental to the paper and currently has been overlooked.

Line by line comments are in the word document attached however specific comments in relation to the SI are below:

- Figure 1 - does the variation in colour have any meaning?
- Table 1 - this should be in the main body of the paper.
- Table 4 - Adult males are listed twice - once with a value of significance and once without.

Overall the paper is very thorough and well written however it feels like some of the fundamental information needs to be provided to enable the reader to reach the same conclusions as the author.

Reviewer #2

(Remarks to the Author)

In this study, "Sea surface temperature and chemical pollution are associated with increased risk of infectious disease mortality in short-beaked common dolphins", the authors investigated for the first time the effect of rising sea surface temperatures in UK waters and chemical pollution, particularly PCBs, on the increase in infectious disease mortality in common dolphins stranded along the UK coast. To do this, the authors used data from a huge dataset of 836 dolphins that were examined post-mortem to determine the cause of death, and for some of them with PCB data. The study covered the period 1990-2020 and then 3 decades. The results of the study showed that there is indeed an effect of both factors (SST and chemical pollution) that increases the risk of infectious disease in this species that frequents UK waters. In addition, the authors presented a new threshold for PCB concentrations of 22 mg/kg lipid weight for common dolphins on the basis of an

increased risk of disease.

In my opinion, this study clearly deserves to be published because the results will make an important contribution to the scientific community in improving our knowledge of the impact of today's major threats, such as chemical pollution and climate change, on marine mammal populations. The study is also well and clearly written, and the context is well presented. However, I have some important points to raise with the authors, which I hope will help to improve this potential publication.

General comments:

- - The main result of the study is that there is an effect of both factors (SST and PCB concentrations, whether they decrease over time) in increasing infectious disease mortality. The GLM used is presented in section 4.3.2. however I don't understand how you obtain a probability of mortality from this model? what is the case in your equation? this model needs to be detailed as it is the crucial part of the study. It is also not clear how the authors arrive at the proposed threshold of 22 mg/kg lw to increase the risk of infectious mortality in this species. Should this be explained in section 4.3.3?
- - This proposed threshold is an important and very useful result, then I suggested (after clarifying how it was obtained) to include it in the abstract.
- - The best and finally selected GLM to investigate the relationship between PCB blubber concentrations, SST and infectious disease mortality is not clearly presented. There are too many tables in the sup info but what we need is the initial model tested and the final model selected with its coefficients and variables that have a significant effect on it. This is a crucial part of the study and in my opinion the authors need to make an effort to clarify and simplify this statistical part. For example, I think it is more important to include this information in the manuscript (rather than in the sup info) than Figure 2, which is not really used in the study.
- As a final general comment, the effect of these factors on disease transmission in common dolphins is quite correct and the results of this study help to reinforce this hypothesis. However, if I am not mistaken, the authors did not take into account that the capacity to detect and identify the cause of death of marine mammals has also developed in the last 3 decades, including in the UK stranding programme, participating to the logical increase of infectious diseases, because the means used to detect the causes of death of marine mammals are very probably better, and perhaps different, than in 1990. According to Table S1, live strandings are the second most common cause of death for common dolphins and more numerous than infectious diseases, have the authors observed whether there is also an increase over time and the possible relationship with the factors tested in the study?

Detailed comments:

I have added comments and some minor corrections to the Word version of the manuscript and supplementary information in an annotated version.

Version 1:

Reviewer comments:

Reviewer #1

(Remarks to the Author)

Thank you for taking the time to diligently address all of the reviewers comments. I have no further comments or concerns and find the manuscript to be significantly improved. Minor edit line 544 you have histology listed twice. Great work and I look forward to seeing it published.

Reviewer #2

(Remarks to the Author)

Comments for author (2nd revision)

I would like to thank the authors of this paper for their consideration of my comments and those of the other reviewer. They greatly improve the manuscript and clarify important parts of it, such as the statistical treatment of the different models carried out, as well as providing more detail on how they calculate the PCB threshold for infectious disease in common dolphins. I also appreciate the expanded explanation of their causes of death and categories, as well as the clarification of the data set used for each purpose of the study (Figure 5). However, I have a number of minor comments, most of which have not been considered or I have not seen in the marked-up version. However, these are not relevant to the publication of the article.

- Finally, I have some minor comments on this second version of the manuscript.

-

- - Again, on the proposed PCB threshold for infectious disease in common dolphins, while how they calculate is clearer to me, I still wonder about the data set for the calculation. Is probably said but not enough and deserves to be highlighted, I propose to add this information in Figure 5 are in section 4.3.3, detailing if they considered or the population cohorts or not (ie adult males, females, subadults ...).

- In Figure 3B, you are sure that you are talking about monthly SST and not annual SST, it is equally important to note this in the figure label and in the legend.

- Line 394. What do you consider to be a control when you talk about PCB concentrations? it is almost impossible to find individuals with no PCB concentrations.

- Line 435. I am very surprised to read that there has been no increase in the number of dolphin deaths in the UK due to by-

catch?

- Line 454. We are in the discussion section, so remove 'Figure 3'.
- Line 647. Are you sure you are referring to Table 1 here?

**Response to Reviewers**

Please see below for our responses to the reviewers' comments and suggestions.

This document has been formatted as outlined below:

Reviewer comment – Normal font black

Authors response – Normal font blue

*Revised text – Italic font blue*

Before we address the reviewers comment below, we wanted to thank the editor for
their specific comments and revision suggestions:

“In particular, please note that the following revisions would be necessary for us to
contact our referees again: clearly defining and classifying infectious diseases (lines
463-486), fully explaining the treatment of data from the original 836 necropsies,
more detail surrounding the GLM and PCB threshold approaches (lines 578-634) ,
discussion of temporal changes in ability to detect cause of death (lines 343-359),
and consideration of SST and PCB effects on live strandings (lines 533-537 and 619-
623).”

Each of these points has been addressed below in our response to reviewers (see
line numbers above), except for the comment regarding the full explanation of the
treatment of the data from the original 836 necropsies, which we want to take the
opportunity to address here.

To make it clearer for the reader which of the datasets have been used in each of the
analyses we have included a new figure within the methods section (Figure 6). We
have referenced this new figure at various points throughout the methods and have
included the following sentence at the start of the “Statistics and Reproducibility”
Section (line 523).

*“Figure 6, located at the end of this section, provides further details on the datasets
used in the models described below, including which variables were present in each
of the subsets.”*

*Figure 6: Schematic depicting the dataset that was used to investigate each question and the associated sample*
 *size.*

**Reviewers' comments:**

Reviewer #1 (Remarks to the Author):

Thank you for the opportunity to review this important manuscript. The major findings
 of the paper are pivotal in marine mammal conservation however there are several
 questions outstanding regarding the methods and approach prior to accepting the
 current conclusions.

Primarily my biggest concern with the claim of association between PCBs, SST and
 increased risk of death from infectious disease is the definition and classification of
 infectious disease. This is fundamental to the conclusion of the paper yet has been
 overlooked throughout the introduction, methods and discussion. I suggest to have
 background information in relation to the infectious diseases of concern in this
 population in the introduction, clear definition as to how cases were classified into
 the infectious disease categories and what these categories were in the methods
 and subsequent discussion as to how these criteria and classification could have
 influenced results. As it stands the reader has to go to the supplementary information
 to look into the table as to what the types of infectious disease are. There is no
 discussion as to the increase in number of cases could have been linked to
 increased reporting or additional diagnostic screening or due to the increased
 number of stranded cases in general (Supplementary Fig. 1A). I suggest that there
 should be additional discussion as to why the number of infectious disease cases
 increased. This is really fundamental to the paper and currently has been
 overlooked.

We are very pleased the reviewer believes our findings are pivotal to marine
 mammal conversation and thank them for their helpful comments and suggestions.
 We have addressed their concerns regarding our methods and approach and believe
 the revisions have greatly enhanced the quality of the paper.

With regards Reviewer 1's concern about the definition and classification of
 infectious disease we have made several changes.

First, we have included background information in relation to infectious diseases in
the introduction (lines 101-117):

*“Determining the threat from stressors such as climate-mediated environmental*
*changes and chemical pollutants is a challenging task. Many studies have reported*
*associations between exposure to PCBs and immunosuppression and/or infectious*
*disease in marine mammals.^{9,24,32} Further, there are several possible mechanisms*
*whereby warming ocean temperatures could increase rates of infectious disease*
*(e.g., increased pathogen transmission rates²² or reduced prey quality and*
*availability compromising immune function). Infectious diseases represent a*
*significant threat to health and survival, with gastritis/enteritis, meningoencephalitis*
*and pneumonia (bacterial, viral and parasitic) being the most prevalent.³³ Gastritis*
*and enteritis, caused by a range of infectious agents, affect the digestive tract,*
*leading to reduced nutrient absorption, dehydration, and, in severe cases, mortality.³⁴*
*Meningoencephalitis, often associated with bacterial or viral pathogens, causes*
*inflammation of the brain and its surrounding membranes, leading to neurological*
*symptoms that can impact navigation, social behaviours and feeding capacity crucial*
*for survival.³⁵ Pneumonia, a respiratory infection that causes inflammation of the air*
*sacs, causes breathing difficulties, which can result in death.^{34,35} While it is not*
*possible to carry out controlled experiments, epidemiological studies can be carried*
*out to investigate associations between infectious disease and exposure to potential*
*stressors such as PCB concentrations and changing ocean temperatures.^{32,36}”*

Second, we have provided clearer definitions as to how cases were classified into
the infectious disease categories and what these categories were. To do so, we have
included the following section in the methods (lines 463-480) and moved Table S1
from the SI to the methods section (line482). We have also included more detail in
this table by splitting the categories into more detailed sub-categories.

*“Each animal had a responsible veterinarian who contemporaneously determined the*
*cause of death by gross examination followed by ancillary testing (including*
*parasitology, bacteriology, histology, virology and histology) as required. ‘Infectious*
*disease’ is a broad category used for analyses within the UK strandings programme,*
*consisting of a number of cause of death categories of infectious origin⁷¹. Within the*
*broad category, we have provided further details on classification and diagnosis for*
*the categories that contained the highest number of cases. Animals were classified*
*as having died from parasitic gastritis and/or enteritis based on the observed extent*
*of parasitism in the area and whether the parasite load was thought to have*
*contributed significantly to the death of the animal (e.g., lack of ability to absorb*
*nutrients). Animals were categorised as having died from meningoencephalitis based*
*on visual inspection of the brain, histopathology of the brain tissue and/or positive*
*culture. Generalised bacterial or fungal infections were diagnosed based on positive*
*culture and histopathological examination of relevant tissues. Animals were classified*
*as having died from pneumonia (from parasitic, bacterial and fungal infections)*
*based on positive culture from sampled sites, histopathology and the extent of lung*
*function impairment derived from visual inspection. In some cases, animals will have*
*comorbidities, therefore the veterinary pathologist will assess which were the most*

*significant in relation to cause of death and the animals will be assigned to that*
*category.”*

Reviewer 1 raised concerns regarding the limited discussion on whether the increase
in the number of cases could have been linked to increased reporting or additional
diagnostic screening or due to the increased number of stranded cases in general.
We acknowledge that this was not addressed in the discussion. We agree with
Reviewer 1 that advancements will have improved the capability to detect and
diagnose infectious disease as a cause of death in marine mammals. However,
concurrently there have also been significant advancements in the diagnosis of other
causes of death (e.g., bycatch/trauma cases). These shifts in diagnostic practices
are challenging to incorporate statistically, as it is difficult to estimate the number of
‘missed’ cases from the past. This limitation is an inherent challenge when working
with datasets that span extended periods. However, the value of these long-term
datasets lies in their large sample sizes, which allow us to explore associations that
would otherwise be impossible to detect in long-lived sentinel species. We believe
this is an essential point to acknowledge, and we have therefore amended the
discussion to include the following text (lines 344-359).

*“It is also important to acknowledge that, given the time span of the dataset, the*
*capacity to diagnose infectious diseases in stranded marine mammals has improved.*
*However, similar advances have also been made in diagnosing cases of bycatch and*
*other causes of mortality.⁵⁴ Therefore, while these improvements in diagnostic*
*capacity may contribute in part to the temporal increase in infectious disease cases,*
*we think it is unlikely that the increase in infectious disease cases is solely due to*
*improved detection. Changes in diagnostic capacity may also have altered the*
*potential for diagnosis of certain disease processes and some infectious disease*
*sub-categories. This should not have significantly impacted our results as we did not*
*investigate associations with specific infectious diseases however, this is an*
*important factor to consider if specific diseases are investigated in the future.*
*Additionally, parasites are particularly vulnerable to overdiagnosis as they are*
*grossly visible and relatively easy to detect. A sick animal with high parasite loads*
*may be classified as an infectious death, but there may be underlying toxic,*
*environmental, social, or alternative infectious causes, such as immunosuppressive*
*morbilliviruses, contributing to the condition. Our findings highlight such complexities,*
*demonstrating the importance of a multifactorial approach in understanding disease*
*processes in marine mammals. “*

Line by line comments are in the word document attached however specific
comments in relation to the SI are below:

- Figure 1 - does the variation in colour have any meaning?

The bars were coloured according to the number of strandings per year with lighter
shades of blue denoting larger numbers. Admittedly, we did not include this
information in the caption so it was not clear. Given the colour scale was not
providing any additional information, we have removed the colour gradient from the
figure.

- Table 1 - this should be in the main body of the paper.

We thank the reviewer for this suggestion and have moved this table from the SI to the main body of the paper (Table 2).

- Table 4 - Adult males are listed twice - once with a value of significance and once without.

We thank the reviewer for drawing our attention to this. The second reference to adult males should have said juveniles and we have corrected the table accordingly.

Overall the paper is very thorough and well written however it feels like some of the fundamental information needs to be provided to enable the reader to reach the same conclusions as the author.

We thank Reviewer 1 for their positive and constructive comments. We think the changes we have in light of their comments have greatly strengthened the paper and we hope they have helped to alleviate the reviewer's concerns.

With regards to the minor revisions/comments we have summarised these below.

We thank reviewer 1 for their helpful suggestions to improve Figure 4. We have amended the colours used in Figure 4 and amended how the confidence intervals are displayed to make it easier to read. We have also limited the y-axis range in the figure in the main body and moved the previous figure to the SI (Fig S3).

We thank the reviewer for their suggestion to include discussion on why there was no significant interaction term between SST and PCB concentrations. We have incorporated the following additional text into the discussion (lines 372-376).

"It is also important to consider the lack of a significant interaction term between SST and PCB concentrations and whether this is a true finding or an artefact of the different temporal and biological dynamics of these variables. While SST varies seasonally and can influence short-term physiological responses, PCB accumulation in blubber represents long-term pollutant exposure, potentially masking any immediate relationship with SST."

Reviewer #2 (Remarks to the Author):

In this study, "Sea surface temperature and chemical pollution are associated with increased risk of infectious disease mortality in short-beaked common dolphins", the authors investigated for the first time the effect of rising sea surface temperatures in UK waters and chemical pollution, particularly PCBs, on the increase in infectious disease mortality in common dolphins stranded along the UK coast. To do this, the authors used data from a huge dataset of 836 dolphins that were examined post-mortem to determine the cause of death, and for some of them with PCB data. The study covered the period 1990-2020 and then 3 decades. The results of the study showed that there is indeed an effect of both factors (SST and chemical pollution) that increases the risk of infectious disease in this species that frequents UK waters.

In addition, the authors presented a new threshold for PCB concentrations of 22
227 mg/kg lipid weight for common dolphins on the basis of an increased risk of disease.

In my opinion, this study clearly deserves to be published because the results will
make an important contribution to the scientific community in improving our
knowledge of the impact of today's major threats, such as chemical pollution and
climate change, on marine mammal populations. The study is also well and clearly
written, and the context is well presented. However, I have some important points to
raise with the authors, which I hope will help to improve this potential publication.

We thank reviewer 2 for their helpful and positive comments and have made substantial
revisions to the paper in light of their suggestions. We believe the manuscript is much
stronger as a result.

General comments:

- - The main result of the study is that there is an effect of both factors (SST and PCB
concentrations, whether they decrease over time) in increasing infectious disease
mortality. The GLM used is presented in section 4.3.2. however I don't understand
how you obtain a probability of mortality from this model? what is the case in your
equation? this model needs to be detailed as it is the crucial part of the study. It is
also not clear how the authors arrive at the proposed threshold of 22 mg/kg lw to
increase the risk of infectious mortality in this species. Should this be explained in
section 4.3.3?

We thank the reviewer for their constructive comments on this section. Addressing
this suggestion, we have added additional text within section 4.3.2 to clarify how the
modelling approach was used to estimate the increased risk of mortality from
infectious diseases (lines 578-601). We have also included references to similar
modelling approaches and provided the model equations in this section to further
clarify the model details.

*"We investigated the relationship between PCB blubber concentrations, SST and*
*infectious disease mortality by fitting generalised linear models with binomial*
*distributions and logit link functions (logistic regression). This approach is used to*
*predict binary outcomes, such as relative mortality risk based on various predictor*
*variables. The probability of the outcome can then be transformed into an odds ratio,*
*which represents how a one-unit increase in a predictor variable (e.g., SST and PCB*
*concentration) affects the relative odds of mortality (e.g., risk of death from infectious*
*disease). Cause of death (cases versus controls) was modelled against monthly and*
*anomaly mean SST, PCB concentrations and other selected covariates were used*
*as potential predictors. The potential predictors were selected according to the*
*biological rationale that they could impact cause of death. The variables included in*
*the full model were body condition, age and sex class, latitude, longitude, season*
*and monthly mean SST, monthly SST anomaly, an interaction term between PCB*
*concentrations and monthly mean SST (Equation 2). All explanatory variables were*
*scaled and centred and assessed for multicollinearity using Variance Inflation*
*Factors (VIF). Significant collinearity was found between season and mean SST and*
*SST anomaly. A key aim of this study was to investigate possible relationships*
*between SST and infectious disease mortality, therefore season and SST anomaly*

were removed from the model. SST anomaly was removed rather than monthly
mean SST as the AIC (Akaike's Information Criterion) for the model with monthly
mean SST was lower. We performed a chi-squared test to test for associations
between season and case/controls to ensure monthly mean SST was not just a
proxy for seasonal variations in mortality causes and found no significant differences
(chi-sq. $X^2=5.2491$, $p<0.05$). We tested all possible variable combinations to obtain
several candidate models which were ranked according to their AIC values (Δ
$AIC<2$). Our final prediction model was obtained by averaging the set of plausible
models (Equation 3).”

$$\begin{aligned} 286 \quad \text{Case} &\sim \beta_0 + \beta_1 \text{Nutritional condition} + \beta_3 \text{Latitude} + \beta_4 \text{Longitude} + \beta_5 \text{SST Mean} \\ 287 \quad &* \beta_5 \text{PCB Blubber Concentration} + \beta_6 \text{Age\&Sex Class} + \beta_7 \text{Season} \\ 288 \quad &+ \beta_8 \text{SST Anomaly} \end{aligned}$$

*Equation 2: The full form of the logistic regression model (before the removal of some terms owing to correlation and before*
*terms were excluded by model averaging) used to investigate relationships between SSTs (sea surface temperature), PCB*
*(polychlorinated biphenyl) blubber concentrations and infectious disease mortality.*

$$\begin{aligned} 293 \quad \text{Case} &\sim \beta_0 + \beta_1 \text{Nutritional condition} + \beta_3 \text{Latitude} + \beta_4 \text{Longitude} + \beta_5 \text{SST Mean} \\ 294 \quad &* \beta_5 \text{PCB Blubber Concentration} \end{aligned}$$

*Equation 3: The final form of the logistic regression model (derived from Equation 2 following model averaging) used to*
*investigate relationships between SSTs (sea surface temperature), PCB (polychlorinated biphenyl) blubber concentrations*
*and infectious disease mortality.*

- - This proposed threshold is an important and very useful result, then I suggested
(after clarifying how it was obtained) to include it in the abstract.

We thank the reviewer for their suggestion and have included the following sentence
in the abstract (line 38).

“Additionally, we derived a novel PCB threshold concentration (22 mg/kg lipid)
defined as the level where PCB blubber concentrations are significantly associated
with infectious disease mortality risk.”

We have also included further information in the methods section with regards to
how this threshold was derived (lines 625-635).

“We derived a threshold at which PCB blubber concentrations have a significant
effect on infectious disease mortality by altering the logistic regression model
described in section 4.3.2 (Equation 2). We replaced the continuous PCB blubber
concentrations with a categorical variable that indicated whether animals were above

*or below a certain concentration (Equation 4). We then iteratively adjusted the PCB*
*concentration to identify the threshold at which PCB concentration reached statistical*
*significance ($p < 0.05$) and became associated with infectious disease mortality.”*

$$\text{Case} \sim \beta_0 + \beta_1 \text{Nutritional condition} + \beta_3 \text{Latitude} + \beta_4 \text{Longitude} + \beta_5 \text{SST Mean}$$

$$* \beta_5 \text{PCB_Threshold}$$

*Equation 4: The final form of the logistic regression model used to derive a threshold PCB concentration.*

We have also amended the results section as detailed below (lines 193-201).

*“When we included PCB concentration as a categorical variable, to derive the*
*threshold at which PCB concentration was significantly associated with infectious*
*disease mortality, we found this occurred at a concentration of 22 mg/kg lipid. The*
*model equation follows the same form as Equation 2 and is detailed in the methods*
*(Equation 4). In both cases, PCB concentration and SST were the only two*
*significant predictors. The coefficients of the model are shown in the Table S5.”*

- - The best and finally selected GLM to investigate the relationship between PCB
blubber concentrations, SST and infectious disease mortality is not clearly
presented. There are too many tables in the sup info but what we need is the initial
model tested and the final model selected with its coefficients and variables that
have a significant effect on it. This is a crucial part of the study and in my opinion the
authors need to make an effort to clarify and simplify this statistical part. For
example, I think it is more important to include this information in the manuscript
(rather than in the sup info) than Figure 2, which is not really used in the study.

We thank the reviewer for this suggestion and have made several revisions to clarify
and simplify this section and have moved Figure 2 to the SI (lines 156-177).

In the results (section 2.2), we have now included both the model equation for the
initial model tested and the equation for the final model selected. Additionally, we
have repositioned the table displaying the final GLM to appear directly beneath the
paragraph detailing the results, making it more accessible for reference.

*“We found that PCB blubber concentrations and monthly mean SST are associated*
*with an increased risk of infectious disease mortality in short-beaked common dolphin*
*(Table 2, Figure 3). The equations of the full model that was tested and the final model*
*are detailed below (Equations 1 and 2). We found that the exposure odds ratio for PCB*
*blubber concentrations and infectious disease mortality was 1.016 (1.01-1.02). Hence,*
*for a 1 mg/kg lipid increase in PCB blubber concentrations, there is an increased*
*relative risk of death from infectious disease of 1.6%. For context, we found that the*
*mean PCB blubber concentration was 32.15 mg/kg lipid, which equates to a 51%*
*increase in relative risk compared to an animal with no PCB burden. The odds ratio*
*for monthly mean SST was 1.14 (1.07-1.21), equating to a 14% increase in relative*
*risk of death from infectious disease for each °C increase in SST.*

$Disease\ Case \sim \beta_0 + \beta_1 Nutritional\ condition + \beta_3 Latitude + \beta_4 Longitude$
 $+ \beta_5 SST\ Mean * \beta_5 PCB\ Blubber\ Concentration + \beta_6 Age\&\ Sex\ Class$
 $+ \beta_7 Season + \beta_8 SST\ Anomaly$

*Equation 1: The full form of the logistic regression model (before the removal of some terms owing to correlation*
 *and before terms were excluded by model averaging) used to investigate relationships between SSTs (sea*
 *surface temperature), PCB (polychlorinated biphenyl) blubber concentrations and infectious disease mortality.*

$Disease\ Case \sim \beta_0 + \beta_1 Nutritional\ condition + \beta_3 Latitude + \beta_4 Longitude$
 $+ \beta_5 SST\ Mean * \beta_5 PCB\ Blubber\ Concentration$

*Equation 2: The form of the averaged logistic regression model used to investigate relationships between SSTs*
 *(sea surface temperature), PCB (polychlorinated biphenyl) blubber concentrations and infectious disease*
 *mortality.*

Following the model coefficient table, we have revised the paragraph describing the
 final model to offer more detail on the selection process. Additionally, we have
 included a table in the Supplementary Information (SI) showing the top models (Δ
 $AIC < 2$) that were averaged to derive the final GLM (lines 184-191).

*“After evaluating all possible variable combinations from the full model (Equation 1),*
 *we ranked them by their AIC values. Our final prediction model was obtained by*
 *averaging the highest-ranked models that had a $\Delta AIC < 2$ relative to the model with*
 *the lowest AIC. The final model was based on an average across seven models .*
 *Details on variable combinations, ΔAIC and model weights are shown in Table S2.*
 *The full averaged model included, nutritional condition, latitude, longitude and an*
 *interaction term between monthly mean SST PCB blubber concentration (Equation*
 *2). We found that monthly mean SST and PCB blubber concentrations were the only*
 *two significant terms in the model (Table 1).”*

Additional table in SI:

*Table 1: Variable combinations, ΔAIC values and model weights for the models that were averaged to obtain the*
 *final generalised linear model (GLM) used to investigate relationships between SSTs (sea surface temperature),*
 *PCB (polychlorinated biphenyl) blubber concentrations and infectious disease mortality.*

Age Sex Class	Latitude	Longitude	Nutritional Condition	Mean SST	PCBs	Mean SST * PCBs	ΔAIC	Weight
-	+	-	-	+	+	-	0	0.11
-	-	-	-	+	+	-	0.37	0.092

-	+	-	+	+	+	-	1.42	0.054
-	+	+	-	+	+	-	1.42	0.054
-	-	+	-	+	+	-	1.48	0.052
-	-	-	+	+	+	-	1.58	0.05
-	+	-	-	+	+	+	1.96	0.041

- As a final general comment, the effect of these factors on disease transmission in common dolphins is quite correct and the results of this study help to reinforce this hypothesis. However, if I am not mistaken, the authors did not take into account that the capacity to detect and identify the cause of death of marine mammals has also developed in the last 3 decades, including in the UK stranding programme, participating to the logical increase of infectious diseases, because the means used to detect the causes of death of marine mammals are very probably better, and perhaps different, than in 1990. According to Table S1, live strandings are the second most common cause of death for common dolphins and more numerous than infectious diseases, have the authors observed whether there is also an increase over time and the possible relationship with the factors tested in the study?

We thank the reviewer for raising these important points and have addressed each of them below.

1. Advancement in detection and diagnostic methods to detect infectious disease.

This was also a point that was raised by Reviewer 1 therefore, we have copied our response below:

We acknowledge that advancements will have improved the capability to detect and diagnose infectious disease as a cause of death in marine mammals. These advancements may indeed contribute to the observed increase in infectious disease cases. However, concurrently there have also been significant advancements in the diagnosis of bycatch/trauma cases. These shifts in diagnostic practices are challenging to incorporate statistically, as it is difficult to estimate the number of 'missed' cases from the past. This limitation is an inherent challenge when working with datasets that span extended periods. However, the value of these long-term datasets lies in their large sample sizes, which allow us to explore associations that would otherwise be impossible to detect in long-lived sentinel species. We believe this is an essential point to acknowledge, and we have therefore amended the discussion to include the following text (lines 344-359).

"It is also important to acknowledge that, given the time span of the dataset, the capacity to diagnose infectious diseases in stranded marine mammals has improved. However, similar advances have also been made in diagnosing cases of bycatch.⁵⁴ Therefore, while these improvements in diagnostic capability are likely to have contributed to the temporal increase in infectious disease cases, we think it is

444 *unlikely that the increase in infectious disease cases is solely due to improved*
*detection. Additionally, this research was in part motivated by anecdotal reports from*
*local volunteers, who have observed an increase in disease cases among common*
*dolphins in recent years. “*

2. Have live strandings increased over time and how do these relate to the
factors investigated in this study?

We thank the reviewer for highlighting this important point and have conducted
further analyses in response. Upon investigating whether there has been a temporal
increase in live strandings, we found no significant trend. To carry out this analysis,
we removed six mass stranding events from the dataset so that trends would not be
heavily skewed by a few extreme incidents. For example, in 2008, 26 short-beaked
common dolphins live stranded and subsequently died. Additionally, it is generally
believed that mass strandings are driven primarily by acute stressors (e.g., noise) or
social factors such as a single animal becoming stranded rather than more chronic
stressors such as pollution or changes to sea surface temperature.

We have amended the methods section to include this analysis as detailed below
(lines 530-537).

*“We then tested whether there was a significant temporal trend in the number of*
*strandings for each cause of death, using the dataset of necropsied animals (n=836).*
*Cause of death was categorised into three classes: infectious disease, trauma and*
*others (the latter including starvation and live stranding). We also tested for trends in*
*live strandings separately as this was the second most common cause of death*
*therefore, we wanted to investigate trends separate from the “Others” cause of death*
*category. We removed mass strandings events from the live strandings so that*
*trends were not heavily skewed by a few rare, extreme events (e.g., the mass*
*stranding of 26 short-beaked common dolphins in 2006).”*

We have amended the results to signpost the results of the temporal trend modelling
for other causes of death in the SI (Table 1) (lines 132-137).

*“When we investigated trends in causes of death, using the smaller dataset (n=836),*
*we found a significant increase in the number of stranded short-beaked common*
*dolphin that died from infectious disease across all regions ($\tau = 0.48$, 2-sided p -value*
*=<0.05) (Figure 1A). However, there were no statistically significant temporal trends*
*in the other cause of death categories (Table S1).”*

We have also conducted additional analysis to investigate potential associations
between live strandings, PCB concentrations and SST. To examine these
associations, we performed another logistic regression, with live strandings
representing the cases (n=23) and trauma and other causes of death, excluding
infectious disease, representing the controls (n=83). We applied the same model
equation (Equation 1) as we used to investigate infectious disease and found that
neither PCB concentration nor SST were significant predictors. Latitude was the only
significant predictor, such that live strandings were more likely to occur at higher

latitudes. We have included this additional analysis in the methods, results and
discussion sections of the paper and pasted the relevant sections below.

Methods (lines 620-624)

*“To further validate the conclusions of our model, we also investigated associations*
*between live strandings (the second most common cause of death), PCB*
*concentrations and SSTs. We followed the same approach as described above but*
*excluded infectious disease cases from the analysis. Live strandings were the cases*
*and animals that died from trauma and other causes of death were the controls.”*

Results (lines 198-201)

*“To further validate the above findings, we repeated the analysis but removed*
*infectious disease cases from the model and set live strandings as the cases with*
*trauma and other causes of death as the controls. We found there was no significant*
*association between live strandings, PCBs and SST (Table S7).”*

Discussion (lines 275-281)

*“Further, as short-beaked common dolphin are considered to be a warm-water*
*adapted species³⁹, our findings are not necessarily intuitive and may therefore, be*
*reflective of changes in the ecosystem precipitated by ocean warming and not*
*mechanistically linked. However, the lack of any significant association between live*
*strandings, PCB concentrations and SSTs does suggest that immune impairment*
*related to these two stressors might be a contributing factor. Given the exploratory*
*nature of this study, further investigation is warranted to ascertain causality.”*

Detailed comments:

I have added comments and some minor corrections to the Word version of the
manuscript and supplementary information in an annotated version.

**Response to Reviewers**

Please see below for our responses to the reviewers' comments and suggestions.

This document has been formatted as outlined below:

Reviewer comment – Normal font black

Authors response – Normal font blue

*Revised text – Italic font blue*

**Reviewers' comments:**

Reviewer #1 (Remarks to the Author):

Thank you for taking the time to diligently address all of the reviewers comments. I
have no further comments or concerns and find the manuscript to be significantly
improved. Minor edit line 544 you have histology listed twice. Great work and I look
forward to seeing it published.

*We are very pleased that reviewer 1 believes we have addressed their concerns and
that the manuscript has significantly improved. We have corrected the error on line
544.*

Reviewer #2 (Remarks to the Author):

Comments for author (2nd revision)

I would like to thank the authors of this paper for their consideration of my comments
and those of the other reviewer. They greatly improve the manuscript and clarify
important parts of it, such as the statistical treatment of the different models carried
out, as well as providing more detail on how they calculate the PCB threshold for
infectious disease in common dolphins. I also appreciate the expanded explanation
of their causes of death and categories, as well as the clarification of the data set
used for each purpose of the study (Figure 5). However, I have a number of minor
comments, most of which have not been considered or I have not seen in the
marked-up version. However, these are not relevant to the publication of the article.

*We are very pleased that reviewer 2 believes our changes in response to their
comments have greatly improved the manuscript and clarifies important parts. We
addressed their minor comments below.*

- Finally, I have some minor comments on this second version of the manuscript.

-

- - Again, on the proposed PCB threshold for infectious disease in common dolphins,
while how they calculate is clearer to me, I still wonder about the data set for the

calculation. Is probably said but not enough and deserves to be highlighted, I
propose to add this information in Figure 5 are in section 4.3.3, detailing if they
considered or the population cohorts or not (ie adult males, females, subadults ...).

We have amended Figure 5 as per their suggestion and amended section 4.3.3 as
follows:

*“We replaced the continuous PCB blubber concentrations with a categorical variable*
*that indicated whether animals were above or below a certain concentration*
*(Equation 4). We did not the stratify the data by sex maturity class.”*

- In Figure 3B, you are sure that you are talking about monthly SST and not annual
SST, it is equally important to note this in the figure label and in the legend.

We have amended the figure label to read monthly SST.

- Line 394. What do you consider to be a control when you talk about PCB
concentrations? it is almost impossible to find individuals with no PCB
concentrations.

We were not intending to talk about controls with respect to PCB concentrations but
rather controls being animals that did not die from infectious disease. We have
clarified this in the text as follows:

*“Therefore, while we attempted to select the cases and controls independently of*
*PCB exposure, if there are discrepancies in PCB concentrations between the control*
*group (animals that died from trauma and are hypothesised to have lower*
*concentrations than disease cases) and the general population, this may have*
*skewed odds ratio estimations.”*

- Line 435. I am very surprised to read that there has been no increase in the
number of dolphin deaths in the UK due to by-catch?

We appreciate the reviewer’s comment and agree that this is an interesting finding.
However, given the focus of this paper on infectious diseases, we considered an in-
depth discussion of bycatch trends to be beyond the scope of this manuscript.

That said, there are several possible explanations for the observed pattern. First, we
analysed proportions of causes of death rather than absolute numbers. As a result,
even if there were an increase in the total number of bycatch cases, it may not be
apparent in our proportional analysis due to the concurrent increase in infectious
disease cases.

Second, our analysis spans a 30-year period, during which there have been
significant fluctuations in bycatch numbers. These fluctuations may be influenced by
changes in diagnostic capacity and fisheries interventions. Consequently, increases
and decreases in bycatch numbers over time may effectively cancel each other out
when viewed over the entire study period.

We hope this clarifies our findings and are happy to elaborate further if deemed
necessary.

- Line 454. We are in the discussion section, so remove 'Figure 3'.

We have made this amendment.

- Line 647. Are you sure you are referring to Table 1 here?

We have made this amendment.

Sea surface temperature and chemical pollution are associated with
increased risk of infectious disease mortality in short-beaked common
dolphins (*Delphinus delphis*).

Rosie S. Williams^{*a,b}, David J. Curnick^a, Andrew Baillie^c, Jonathan L. Barber^d, James
Barnett^e, Andrew Brownlow^f, Robert Deaville^a, Nicholas J. Davison^f, Mariel ten Doeschate^f,
Paul D Jepson, Sinéad Murphy^g, Rod Penrose^h, Matthew Perkins^a, Simon Spiro^a, Ruth
Williamsⁱ, Michael J. Williamson^a, Andrew A. Cunningham^a, Andrew C. Johnson^j

10 ^a Institute of Zoology, Zoological Society of London, Regent's Park, London NW1 4RY, United Kingdom

11 ^b Department of Genetics, Evolution and Environment, University College London, Darwin Building, 99-105
12 Gower Street, London WC1E 6BT, United Kingdom

13 ^c The Natural History Museum, Cromwell Road, London SW7 5BD, United Kingdom

14 ^d Centre for Environment, Fisheries and Aquaculture Science (Cefas), Lowestoft, Suffolk, NR33 0HT, UK

15 ^e Cornwall Marine Pathology Team, Fishers Well, Higher Brill, Constantine, Falmouth, Cornwall TR11 5QG,
16 United Kingdom

17 ^f School of Biodiversity, One Health and Veterinary Medicine, College of Medical, Veterinary & Life Sciences
18 University of Glasgow, Glasgow, G12 8QQ

19 ^g Marine and Freshwater Research Centre, Department of Natural Sciences, School of Science and Computing,
20 Atlantic Technological University, Galway, Ireland.

21 ^h Marine Environmental Monitoring, Penwalk, Llechryd, Cardigan, Ceredigion SA43 2PS, United Kingdom

22 ⁱ Cornwall Wildlife Trust, Truro, Cornwall TR4 9DJ, United Kingdom

23 ^j UK Centre for Ecology and Hydrology, Wallingford, OX10 8BB, UK

**Abstract**

The concurrent pressures of climate change and chemical pollution, often studied in isolation,
have been linked to increases in infectious disease that threaten biodiversity. Understanding
their interconnected nature is vital, as the impacts of climate-mediated environmental changes
can be exacerbated by chemical pollution and vice versa. Using data from 836 UK-stranded
short-beaked common dolphins (*Delphinus delphis*) (n=153 with polychlorinated biphenyl
(PCB) blubber concentrations) necropsied between 1990-2020, we show that PCB
concentrations and sea surface temperatures (SSTs) are associated with an increased risk of
infectious disease mortality. Specifically, a 1 mg/kg lipid increase in PCB concentration
correlates with a 1.6% increase in disease mortality risk, while a 1°C rise in SST corresponds
to a 14% increase. International efforts to reduce carbon emissions have mostly failed and
despite regulatory efforts, PCBs remain a significant threat. Our findings demonstrate the
urgent need for conservation strategies that address both risk factors simultaneously to protect
marine biodiversity.

Keywords: climate change; pollution; polychlorinated biphenyls; common dolphins; disease;
marine mammals; sea surface temperature; strandings; cetaceans

**1. Introduction**

Climate change and chemical pollution have both been linked to increases in infectious disease
that pose a significant threat to biodiversity¹⁻⁴, especially in marine environments that often
serve as the final sink for persistent chemical pollutants (POPs)⁵. The cumulative impacts of
these stressors are likely to produce compounding effects. For example, POPs, such as
polychlorinated biphenyls (PCBs), cause immunosuppression, which may be exacerbated by
climate-mediated environmental changes⁶. Conversely, chemical pollutants may increase the
susceptibility of species to the impacts of climate change⁶⁻⁸. Advancing our knowledge of how
these stressors relate to infectious disease is vital to the development of effective conservation
strategies.

PCBs, which are particularly persistent and toxic in comparison to other POPs, are of particular
concern as they impair reproduction and produce immunotoxicity in marine mammals⁹⁻¹¹.
Despite the European ban on PCBs in the mid-1980s large amounts still require disposal and
they continue to persist in the environment due to their high stability and bioaccumulative
nature^{12,13}. Further, legacy PCBs continue to enter the marine environment via several
mechanisms such as terrestrial run off, inadequate waste disposal, intentional discharge,
dredging, atmospheric transport, dispersion and deposition^{14,15}. However, despite the general
adherence to international regulatory agreements like the Stockholm Convention, tissue
concentrations of PCBs remain at hazardous levels in many wildlife species^{13,16}.

Concurrently, climate change is precipitating rapid alterations to the marine environment,
including increases in the frequency and severity of marine heatwaves¹⁷. In the UK, there have
been increases in mean sea surface temperatures and several marine heatwaves¹⁸⁻²⁰. These

changes are having profound effects on marine ecosystems, including shifts in species
distribution, changes to prey availability and altered pathogen-host dynamics²¹. Further,
warming ocean temperatures have been linked to greater incidence and spread of diseases in
marine species, potentially amplifying the immunosuppressive impacts of chemical pollutants
like PCBs^{6,22}.

Marine mammals are excellent sentinels for marine ecosystem health owing to their high
trophic position, long lifespan and thick layer of blubber where lipophilic pollutants
accumulate²³. They accumulate some of the highest recorded concentrations of PCBs in
wildlife, and exposure to high concentrations has been associated with immune system
impairment in *in vivo* and *in vitro* studies^{9,24}. In the UK, short-beaked common dolphins
(*Delphinus delphis*) are exposed to high levels of PCBs and shifts in their distribution have
been associated with changes in sea surface temperature (SST)^{16,20,25}. Moreover, the rate of
decline of PCBs in the blubber of UK-stranded short-beaked common dolphins is slower
compared to other odontocete species, suggesting an elevated risk of exposure and adverse
effects^{16,26}. In addition, shifts from pelagic to coastal waters may be occurring, as evidenced
by increased near shore sightings²⁷⁻²⁹. This may reflect a response to climate-induced changes
in prey availability, while also increasing exposure to PCBs and other pollutants as coastal
biota often have higher pollutant burdens³⁰. Previous changes in short-beaked common dolphin
distribution in the UK have been linked to changes in the distribution of the European flying
squid (*Todarodes sagittatus*), an important prey species³¹. These changes demonstrate the
intricate relationship between climate-mediated environmental changes and pollutant
exposure. Considering the impacts of these pressures together not only has local relevance but
also global implications, given similar threats faced by marine mammals in industrialised
coastal regions worldwide.

Using data obtained from necropsies of stranded short-beaked common dolphins over three
decades (1990-2020), we sought to understand the cumulative impacts of chemical pollution
and climate-induced changes on infectious disease susceptibility in this species. By
investigating temporal trends in infectious disease mortality and the possible associations
between PCB blubber concentrations, SST and infectious disease mortality, we provide
insights into the broader implications of these environmental stressors on the health of this
sentinel species. In addition, we determine temporal trends in PCB blubber concentrations and
derive a novel infectious disease mortality risk threshold for PCBs in marine mammals.

**2. Results**

**2.1 Temporal trends in causes of mortality**

When we analysed temporal trends in overall strandings (n=3197), we found that between 1990
and 2020 there was a significant increase in the overall number of strandings in the UK ($\tau =$
0.56, 2-sided p-value <0.05) (Figure S1). When we investigated trends in causes of death, using
the smaller dataset (n=836), we found a significant increase in the number of stranded short-
beaked common dolphin that died from infectious disease across all regions ($\tau = 0.48$, 2-sided
p-value = <0.05) (Figure 1A). However, there were no statistically significant temporal trends
in the other cause of death categories. We were unable to test trends in specific types of
infectious disease due to the small sizes in each category. We did plot the absolute number of
cases and yearly frequencies in each category and visual inspection showed no single category
drove the proportional increase in cases (Figure 2). When we incorporated the Oslo Paris
Convention (OSPAR) region classifications (either two or four geographical regions) into our
analysis, we did not find any regional differences in the temporal trends of mortality causes.
Nonetheless, visual representations of these data over each decade show an increase in the

proportion of deaths ascribed to infectious disease in all regions (Figure 1B). For both the
 regional and UK-wide models, the negative binomial regression provided a better fit compared
 to the *glm* with a Poisson distribution.

*Figure 1: (A) Yearly numbers of common dolphin infectious disease cases diagnosed at necropsy (B) Regional proportions*
 *for each cause of death category for each decade. The figures within each bar indicate the percentage for each cause of*
 *death category.*

*Figure 2: (A) Number of cases in each infectious disease category (B) Yearly frequencies for each infectious disease*
 *category for short-beaked common dolphin that underwent necropsy.*

**2.2 Drivers of infectious disease mortality**

We found that PCB blubber concentrations and monthly mean SST are associated with an
increased risk of infectious disease mortality in short-beaked common dolphin (Table 2, Figure
3). When SST was calculated across the four OSPAR sub-regions, rather than the two OSPAR
regions, the model fit improved. For the model using SST derived across four regions, we found
that the exposure odds ratio for PCB blubber concentrations and infectious disease mortality
was 1.016 (1.01-1.02). Hence, for a 1 mg/kg lipid increase in PCB blubber concentrations,
there is an increased relative risk of death from infectious disease of 1.6%. For context, we
found that the mean PCB blubber concentration was 32.15 mg/kg lipid, which equates to 51%
increase in relative risk. The odds ratio for monthly mean SST was 1.14 (1.07-1.21), equating
to a 14% increase in relative risk of death from infectious disease for each °C increase in SST.

*Figure 3: The blue line represents the probability of infectious disease mortality against (A) the sum of 25 chlorobiphenyl*
 *congeners ($\sum 25$ CBs (mg kg⁻¹ lipid)) blubber concentrations at the minimum and maximum monthly mean SSTs in °C (B)*
 *Monthly Mean Sea Surface Temperature (SST) when the regions were divided in four and PCB concentration was set at the*
 *minimum and mean concentration. The shaded blue areas represent the 95% confidence intervals. The mean value has been*
 *chosen for other predictors.*

$$\text{Case} \sim \beta_0 + \beta_1 \text{Nutritional condition} + \beta_3 \text{Latitude} + \beta_4 \text{Longitude} + \beta_5 \text{SST Mean}$$

$$* \beta_5 \text{PCB Blubber Concentration}$$

*Equation 1: The final form of the logistic regression model used to investigate relationships between SSTs (sea surface*
 *temperature), PCB (polychlorinated biphenyl) blubber concentrations and infectious disease mortality.*

The averaged model included, nutritional condition, latitude, longitude and an interaction term
 between monthly mean SST PCB blubber concentration (Equation 1). We found that monthly
 mean SST and PCB blubber concentrations were the only two significant terms in the model

(Table 2). We found the threshold concentration at which PCB blubber concentrations had a
 significant impact on infectious disease mortality risk was 22 mg/kg lipid. The coefficients of
 the model used to derive the threshold are shown in the Table S6.

The results of the model with the larger dataset (n=836), that did not include PCB
 concentration, were similar in that infectious disease mortality risk was positively associated
 with SST and trauma and other causes of death were negatively associated with SST. The
 model included age and sex class, latitude, longitude, nutritional condition and SST but only
 SST and latitude were significant terms. The model coefficients are shown in SI Tables 2&3.

*Table 2: Model averaged coefficients for SST derived for four OSPAR areas, variables were centred and scaled. The*
 *averaged coefficients for SST derived for two OSPAR areas are shown in Table S2.*

	Estimate	Std. Error	Adjusted SE	z value	Pr(> z)
(Intercept)	-6.66	4.61	4.63	1.44	0.15
Latitude	6.18	4.12	4.16	1.49	0.14
Monthly mean SST	1.55	0.77	0.78	1.99	<0.05*
PCB concentration	0.02	0.01	0.01	2.11	<0.05*
Nutritional Condition	-0.19	0.21	0.22	0.87	0.38
Longitude	0.14	0.16	0.02	0.18	0.86
Monthly mean SST: PCB					
concentration	0.00	0.01	0.02	0.18	0.86

**2.3 PCB Blubber Concentration and SST Temporal Trends**

Modelled PCB blubber concentrations decreased between 1999-2020 but the rate of decline
 appears slow and concentrations in adult males are still above the widely used threshold for
 physiological effects and the threshold we derived for increased risk of infectious disease

mortality (Figure 4A). The final model included date of stranding as the response variable with
cause of death, latitude, longitude and age and sex class as explanatory variables (Table S6).
When we subset the data by age and sex class we found there was no statistically significant
downward trend in PCB blubber concentrations in adult males. However, there was a
significant decline in PCB concentrations in adult females and juveniles (SI Tables 7-9).

When we modelled monthly mean SST we found that SSTs have significantly increased in two
of the four OSPAR sub-regions when modelled separately (OSPIIN estimate = 9.026×10^{-5} , p

[revised manuscript text omitted]

6 Barnett^e, Andrew Brownlow^f, Robert Deaville^a, Nicholas J. Davison^f, Mariel ten Doeschate^f,
7 Paul D Jepson, Sinéad Murphy^g, Rod Penrose^h, Matthew Perkins^a, Simon Spiro^a, Ruth
8 Williamsⁱ, Michael J. Williamson^a, Andrew A. Cunningham^a, Andrew C. Johnson^j

9

10 ^a Institute of Zoology, Zoological Society of London, Regent's Park, London NW1 4RY, United Kingdom

11 ^b Department of Genetics, Evolution and Environment, University College London, Darwin Building, 99-105
12 Gower Street, London WC1E 6BT, United Kingdom

13 ^c The Natural History Museum, Cromwell Road, London SW7 5BD, United Kingdom

14 ^d Centre for Environment, Fisheries and Aquaculture Science (Cefas), Lowestoft, Suffolk, NR33 0HT, UK

15 ^e Cornwall Marine Pathology Team, Fishers Well, Higher Brill, Constantine, Falmouth, Cornwall TR11 5QG,
16 United Kingdom

17 ^f School of Biodiversity, One Health and Veterinary Medicine, College of Medical, Veterinary & Life Sciences
18 University of Glasgow, Glasgow, G12 8QQ

19 ^g Marine and Freshwater Research Centre, Department of Natural Sciences, School of Science and Computing,
20 Atlantic Technological University, Galway, Ireland.

21 ^h Marine Environmental Monitoring, Penwalk, Llechryd, Cardigan, Ceredigion SA43 2PS, United Kingdom

22 ⁱ Cornwall Wildlife Trust, Truro, Cornwall TR4 9DJ, United Kingdom

23 ^j UK Centre for Ecology and Hydrology, Wallingford, OX10 8BB, UK

**Abstract**

The concurrent pressures of climate change and chemical pollution, often studied in isolation,
have been linked to increases in infectious disease that threaten biodiversity. Understanding
their interconnected nature is vital, as the impacts of climate-mediated environmental changes
can be exacerbated by chemical pollution and vice versa. Using data from 836 UK-stranded
short-beaked common dolphins (n=153 with polychlorinated biphenyl (PCB) blubber
concentrations) necropsied between 1990-2020, we show that PCB concentrations and sea
surface temperatures (SSTs) are associated with an increased risk of infectious disease
mortality. Specifically, a 1 mg/kg lipid increase in PCB concentration correlates with a 1.6%
increase in disease mortality risk, while a 1°C rise in SST corresponds to a 14% increase.
International efforts to reduce carbon emissions have mostly failed and despite regulatory
efforts, PCBs remain a significant threat. Our findings demonstrate the urgent need for
conservation strategies that address both risk factors simultaneously to protect marine
biodiversity.

Keywords: climate change; pollution; polychlorinated biphenyls; common dolphins; disease;
marine mammals; sea surface temperature; strandings; cetaceans

**1. Introduction**

Climate change and chemical pollution have both been linked to increases in infectious disease
that pose a significant threat to biodiversity¹⁻⁴, especially in marine environments that often
serve as the final sink for persistent chemical pollutants (POPs)⁵. The cumulative impacts of
these stressors are likely to produce compounding effects. For example, POPs, such as
polychlorinated biphenyls (PCBs), cause immunosuppression, which may be exacerbated by
climate-mediated environmental changes⁶. Conversely, chemical pollutants may increase the
susceptibility of species to the impacts of climate change⁶⁻⁸. Advancing our knowledge of how
these stressors relate to infectious disease is vital to the development of effective conservation
strategies.

PCBs, which are particularly persistent and toxic in comparison to other POPs, are of particular
concern as they impair reproduction and produce immunotoxicity in marine mammals⁹⁻¹¹.
Despite the European ban on PCBs in the mid-1980s large amounts still require disposal and
they continue to persist in the environment due to their high stability and bioaccumulative
nature^{12,13}. Further, legacy PCBs continue to enter the marine environment via several
mechanisms such as terrestrial run off, inadequate waste disposal, intentional discharge,
dredging, atmospheric transport, dispersion and deposition^{14,15}. However, despite the general
adherence to international regulatory agreements like the Stockholm Convention, tissue
concentrations of PCBs remain at hazardous levels in many wildlife species^{13,16}.

Concurrently, climate change is precipitating rapid alterations to the marine environment,
including increases in the frequency and severity of marine heatwaves¹⁷. In the UK, there have
been increases in mean sea surface temperatures and several marine heatwaves¹⁸⁻²⁰. These

changes are having profound effects on marine ecosystems, including shifts in species
distribution, changes to prey availability and altered pathogen-host dynamics²¹. Further,
warming ocean temperatures have been linked to greater incidence and spread of diseases in
marine species, potentially amplifying the immunosuppressive impacts of chemical pollutants
like PCBs^{6,22}.

Marine mammals are excellent sentinels for marine ecosystem health owing to their high
trophic position, long lifespan and thick layer of blubber where lipophilic pollutants
accumulate²³. They accumulate some of the highest recorded concentrations of PCBs in
wildlife, and exposure to high concentrations has been associated with immune system
impairment in *in vivo* and *in vitro* studies^{9,24}. In the UK, short-beaked common dolphins
(*Delphinus delphis*) are exposed to high levels of PCBs and shifts in their distribution have
been associated with changes in sea surface temperature (SST)^{16,20,25}. Moreover, the rate of
decline of PCBs in the blubber of UK-stranded short-beaked common dolphins is slower
compared to other odontocete species, suggesting an elevated risk of exposure and adverse
effects^{16,26}. In addition, shifts from pelagic to coastal waters may be occurring, as evidenced
by increased near shore sightings²⁷⁻²⁹. This may reflect a response to climate-induced changes
in prey availability, while also increasing exposure to PCBs and other pollutants as coastal
biota often have higher pollutant burdens³⁰. Previous changes in short-beaked common dolphin
distribution in the UK have been linked to changes in the distribution of the European flying
squid (*Todarodes sagittatus*), an important prey species³¹. These changes demonstrate the
intricate relationship between climate-mediated environmental changes and pollutant
exposure. Considering the impacts of these pressures together not only has local relevance but
also global implications, given similar threats faced by marine mammals in industrialised
coastal regions worldwide.

Using data obtained from necropsies of stranded short-beaked common dolphins over three
decades (1990-2020), we sought to understand the cumulative impacts of chemical pollution
and climate-induced changes on infectious disease susceptibility in this species. By
investigating temporal trends in infectious disease mortality and the possible associations
between PCB blubber concentrations, SST and infectious disease mortality, we provide
insights into the broader implications of these environmental stressors on the health of this
sentinel species. In addition, we determine temporal trends in PCB blubber concentrations and
derive a novel infectious disease mortality risk threshold for PCBs in marine mammals.

**2. Results**

**2.1 Temporal trends in causes of mortality**

When we analysed temporal trends in overall strandings (n=3197), we found that between 1990
and 2020 there was a significant increase in the overall number of strandings in the UK ($\tau =$
0.56, 2-sided p-value <0.05) (Figure S1). When we investigated trends in causes of death, using
the smaller dataset (n=836), we found a significant increase in the number of stranded short-
beaked common dolphin that died from infectious disease across all regions ($\tau = 0.48$, 2-sided
p-value = <0.05) (Figure 1A). However, there were no statistically significant temporal trends
in the other cause of death categories. We were unable to test trends in specific types of
infectious disease due to the small sizes in each category. We did plot the absolute number of
cases and yearly frequencies in each category and visual inspection showed no single category
drove the proportional increase in cases (Figure 2). When we incorporated the Oslo Paris
Convention (OSPAR) region classifications (either two or four geographical regions) into our
analysis, we did not find any regional differences in the temporal trends of mortality causes.
Nonetheless, visual representations of these data over each decade show an increase in the

proportion of deaths ascribed to infectious disease in all regions (Figure 1B). For both the
 regional and UK-wide models, the negative binomial regression provided a better fit compared
 to the *glm* with a Poisson distribution.

*Figure 1: (A) Yearly numbers of common dolphin infectious disease cases diagnosed at necropsy (B) Regional proportions*
 *for each cause of death category for each decade. The figures within each bar indicate the percentage for each cause of*
 *death category.*

*Figure 2: (A) Number of cases in each infectious disease category (B) Yearly frequencies for each infectious disease*
 *category for short-beaked common dolphin that underwent necropsy.*

**2.2 Drivers of infectious disease mortality**

We found that PCB blubber concentrations and monthly mean SST are associated with an
increased risk of infectious disease mortality in short-beaked common dolphin (Table 2, Figure
3). When SST was calculated across the four OSPAR sub-regions, rather than the two OSPAR
regions, the model fit improved. For the model using SST derived across four regions, we found
that the exposure odds ratio for PCB blubber concentrations and infectious disease mortality
was 1.016 (1.01-1.02). Hence, for a 1 mg/kg lipid increase in PCB blubber concentrations,
there is an increased relative risk of death from infectious disease of 1.6%. For context, we
found that the mean PCB blubber concentration was 32.15 mg/kg lipid, which equates to 51%
increase in relative risk. The odds ratio for monthly mean SST was 1.14 (1.07-1.21), equating
to a 14% increase in relative risk of death from infectious disease for each °C increase in SST.

*Figure 3: The blue line represents the probability of infectious disease mortality against (A) the sum of 25 chlorobiphenyl*
 *congeners ($\sum 25$ CBs (mg kg⁻¹ lipid)) blubber concentrations at the minimum and maximum monthly mean SSTs in °C (B)*
 *Monthly Mean Sea Surface Temperature (SST) when the regions were divided in four and PCB concentration was set at the*
 *minimum and mean concentration. The shaded blue areas represent the 95% confidence intervals. The mean value has been*
 *chosen for other predictors.*

$$\text{Case} \sim \beta_0 + \beta_1 \text{Nutritional condition} + \beta_3 \text{Latitude} + \beta_4 \text{Longitude} + \beta_5 \text{SST Mean}$$

$$* \beta_5 \text{PCB Blubber Concentratio}$$

*Equation 1: The final form of the logistic regression model used to investigate relationships between SSTs (sea surface*
 *temperature), PCB (polychlorinated biphenyl) blubber concentrations and infectious disease mortality.*

The averaged model included, nutritional condition, latitude, longitude and an interaction term
 between monthly mean SST PCB blubber concentration (Equation 1). We found that monthly
 mean SST and PCB blubber concentrations were the only two significant terms in the model

(Table 2). We found the threshold concentration at which PCB blubber concentrations had a
 significant impact on infectious disease mortality risk was 22 mg/kg lipid. The coefficients of
 the model used to derive the threshold are shown in the Table S6.

The results of the model with the larger dataset (n=836), that did not include PCB
 concentration, were similar in that infectious disease mortality risk was positively associated
 with SST and trauma and other causes of death were negatively associated with SST. The
 model included age and sex class, latitude, longitude, nutritional condition and SST but only
 SST and latitude were significant terms. The model coefficients are shown in SI Tables 2&3.

*Table 2: Model averaged coefficients for SST derived for four OSPAR areas, variables were centred and scaled. The*
 *averaged coefficients for SST derived for two OSPAR areas are shown in Table S2.*

	Estimate	Std. Error	Adjusted SE	z value	Pr(> z)
(Intercept)	-6.66	4.61	4.63	1.44	0.15
Latitude	6.18	4.12	4.16	1.49	0.14
Monthly mean SST	1.55	0.77	0.78	1.99	<0.05*
PCB concentration	0.02	0.01	0.01	2.11	<0.05*
Nutritional Condition	-0.19	0.21	0.22	0.87	0.38
Longitude	0.14	0.16	0.02	0.18	0.86
Monthly mean SST: PCB					
concentration	0.00	0.01	0.02	0.18	0.86

**2.3 PCB Blubber Concentration and SST Temporal Trends**

Modelled PCB blubber concentrations decreased between 1999-2020 but the rate of decline
 appears slow and concentrations in adult males are still above the widely used threshold for
 physiological effects and the threshold we derived for increased risk of infectious disease

mortality (Figure 4A). The final model included date of stranding as the response variable with
cause of death, latitude, longitude and age and sex class as explanatory variables (Table S6).
When we subset the data by age and sex class we found there was no statistically significant
downward trend in PCB blubber concentrations in adult males. However, there was a
significant decline in PCB concentrations in adult females and juveniles (SI Tables 7-9).

When we modelled monthly mean SST we found that SSTs have significantly increased in two
of the four OSPAR sub-regions when modelled separately (OSPIIN estimate = 9.026×10^{-5} , p

[revised manuscript text omitted]

Figures

Figure 1: Year totals of the overall number of reported strandings for common dolphins

Tables

Table 1: Cause of death class and category for all common dolphins that underwent necropsy

Cause of Death Class	Cause of Death Category	n
Infectious Disease	Gastritis and/or Enteritis	29
Infectious Disease	Others	24
Infectious Disease	Generalised Infection, Bacterial	18
Infectious Disease	(Meningo)encephalitis	17
Infectious Disease	Pneumonia	25
Infectious Disease	Generalised Infection, Fungal	1
Others	Live Stranding	151
Others	Not Established	99
Others	Starvation	42
Others	Others	31

Others	Gas Embolism	3
Others	Pneumonia, Non-Infectious	3
Trauma	Bycatch	346
Trauma	Physical Trauma, Uncategorised	23
Trauma	Physical Trauma, Boat/Ship Strike	12
Trauma	Physical Trauma, Bottlenose Dolphin Attack	6

Table 2: Model averaged coefficients for SST derived for two OSPAR areas, variables were centered and scaled.

	Estimate	Std. Error	Adjusted SE	z value	Pr(> z)
(Intercept)	-6.28	4.71	4.73	1.33	0.18
Latitude	3.42	4.33	4.35	0.79	0.43
Monthly mean SST	1.06	0.95	0.96	1.10	0.27
PCB concentration	0.02	0.01	0.01	2.36	<0.05*
Nutritional Condition	-0.06	0.15	0.15	0.42	0.68
Longitude	0.00	0.05	0.05	0.10	0.92
Monthly mean SST :PCB concentration	0.00	0.01	0.01	0.05	0.96

Table 3: Results of model with larger dataset that did not include PCB concentration with SSTs calculated using two OSPAR areas. Variables were centered and scaled.

	Estimate	Std. Error	Adjusted SE	z value	Pr(> z)
(Intercept)	-0.61	0.50	0.50	1.23	0.22
Adult Males	-0.26	0.37	0.37	0.71	0.48
Juvenile	-0.50	0.49	0.49	1.02	0.31
Latitude	1.04	0.17	0.17	6.01	<0.05*
Nutritional Condition	-0.12	0.18	0.18	0.65	0.51
Monthly mean SST	0.84	0.14	0.14	6.02	<0.05*
Longitude	0.03	0.08	0.08	0.40	0.69

Table 4: Results of model with larger dataset that did not include PCB concentration with SSTs calculated using four OSPAR areas. Variables were centered and scaled.

	Estimate	Std. Error	Adjusted SE	z value	Pr(> z)
(Intercept)	-24.129	10.779	10.808	2.233	0.026
Adult Males	-0.508	0.399	0.400	1.271	0.204
Juveniles	-0.966	0.391	0.392	2.462	0.014
Latitude	0.427	0.210	0.211	2.024	0.043
Nutritional Condition	1.172	1.994	1.999	0.587	0.557

Monthly mean SST	-0.850	0.955	0.957	0.889	0.374
Adult Males	0.304	0.047	0.047	6.452	<0.05*
Latitude:Longitude	-0.017	0.039	0.039	0.439	0.661

Table 5: Model averaged coefficients for SST derived for two OSPAR areas, to derive PCB threshold for significance.

(Intercept)	-48.16	11.73	-4.10	<0.05*
PCB Threshold	1.66	0.70	2.38	<0.05*
Adult Males	-0.92	0.87	-1.06	0.29
Juvenile	-2.06	0.83	-2.48	0.01*
Latitude	0.86	0.22	3.87	<0.05*
Longitude	-0.08	0.22	-0.34	0.73
Monthly mean SST	0.34	0.11	3.10	<0.05*
Nutritional Condition	-6.31	2.31	-2.73	<0.05*

Table 6: Model averaged coefficients for temporal trend of PCBs.

	Estimate	Std. Error	Adjusted SE	z value	Pr(> z)
(Intercept)	-47.02	11.24	11.40	4.13	<0.05*
Adult Males	-2.05	1.15	1.17	1.76	0.08
Juvenile	-2.65	0.95	0.96	2.76	<0.05*
Latitude	43.91	11.01	11.17	3.93	<0.05*
Nutritional Condition	-1.06	0.36	0.37	2.89	<0.05*
Monthly mean SST	3.86	1.22	1.24	3.11	<0.05*
PCB Concentration	1.70	0.71	0.72	2.36	<0.05*
Longitude	0.06	0.16	0.16	0.38	0.71

Table 7: Model averaged coefficients for temporal trend of PCBs in adult males

	Estimate	Std. Error	Adjusted SE	z value	Pr(> z)
(Intercept)	5.22	2.38	2.44	2.14	<0.05*
Date.Found	0.00	0.00	0.00	1.50	0.13
Latitude	-0.04	0.07	0.07	0.57	0.57
Rel.body.wt	-0.53	1.11	1.15	0.46	0.64
Longitude	0.04	0.16	0.17	0.23	0.82

Table 8: Model averaged coefficients for temporal trend of PCBs in adult females

	Estimate	Std. Error	Adjusted SE	z value	Pr(> z)
--	----------	------------	-------------	---------	----------

(Intercept)	11.57	1.68	1.72	6.74	<0.05*
Date.Found	0.00	0.00	0.00	2.95	<0.05*
Latitude	-0.15	0.03	0.03	4.52	<0.05*
Rel.body.wt	-0.45	0.55	0.56	0.81	0.42
Longitude	0.03	0.36	0.37	0.09	0.93
Latitude:Longitude	0.00	0.02	0.02	0.01	1.00

Table 9: Model averaged coefficients for temporal trend of PCBs in juveniles

	Estimate	Std. Error	Adjusted SE	z value	Pr(> z)
(Intercept)	17.64	13.10	13.46	1.31	0.19
Date.Found	0.00	0.00	0.00	2.01	<0.05*
Latitude	-0.27	0.26	0.27	1.01	0.31
Rel.body.wt	-2.63	1.19	1.22	2.15	<0.05*
Longitude	-0.39	3.85	3.95	0.10	0.92
Latitude:Longitude	0.05	0.15	0.16	0.31	0.76

Table 10: Percentage of non-detects for each congener (n=217)

Percentage of non-detects	Congener
35.94	CB.18
35.02	CB.28
49.31	CB.31
8.76	CB.44
3.69	CB.47
11.98	CB.49
1.38	CB.52
1.84	CB.66
0.00	CB.101
1.84	CB.105
4.61	CB.110
0.00	CB.118
0.92	CB.128
0.00	CB.138
7.83	CB.141
0.00	CB.149
0.92	CB.151
0.00	CB.153
1.84	CB.156
3.69	CB.158
0.00	CB.170
0.00	CB.180
0.92	CB.183

Percentage of non-detects	Congener
0.00	CB.187
0.92	CB.194